# Role of competition between polarity sites in establishing a unique front

Chi-Fang Wu[1], Jian-Geng Chiou[1], Maria Minakova[2], Benjamin Woods[1], Denis Tsygankov[2], Trevin R Zyla[1], Natasha S Savage[3], Timothy C Elston[2], Daniel J Lew[1]*

[1]Department of Pharmacology and Cancer Biology, Duke University Medical Center, Durham, United States; [2]Department of Pharmacology, University of North Carolina at Chapel Hill, Chapel Hill, United States; [3]Institute of Integrative Biology, University of Liverpool, Liverpool, United Kingdom

**Abstract** Polarity establishment in many cells is thought to occur via positive feedback that reinforces even tiny asymmetries in polarity protein distribution. Cdc42 and related GTPases are activated and accumulate in a patch of the cortex that defines the front of the cell. Positive feedback enables spontaneous polarization triggered by stochastic fluctuations, but as such fluctuations can occur at multiple locations, how do cells ensure that they make only one front? In polarizing cells of the model yeast *Saccharomyces cerevisiae*, positive feedback can trigger growth of several Cdc42 clusters at the same time, but this multi-cluster stage rapidly evolves to a single-cluster state, which then promotes bud emergence. By manipulating polarity protein dynamics, we show that resolution of multi-cluster intermediates occurs through a greedy competition between clusters to recruit and retain polarity proteins from a shared intracellular pool.

## Introduction

Differentiated cells exhibit a stunning variety of morphologies that enable specialized cell-specific functions. Morphological diversity emerges, in part, from specialization of cortical domains, which are often demarcated by the local accumulation of active GTPases. Among the best-understood cortical specification events is the establishment of cell polarity, wherein local accumulation of a cortical Rho-family GTPase (Cdc42, Rac, or Rop depending on the organism) creates a region destined to become the 'front' (*Etienne-Manneville, 2004*; *Park and Bi, 2007*; *Yang and Lavagi, 2012*). For some cells, restricting polarity to a single front is absolutely imperative: for example, a migrating leukocyte with two fronts would split itself apart (*Houk et al., 2012*). However, other cells routinely specify more than one front: for example, neurons can grow several neurites simultaneously, each with a front-like tip (*Dotti et al., 1988*). Similar phenomena occur in plants and fungi, raising the question of how different cell types generate the correct number of fronts (*Wu and Lew, 2013*). Here we focus on the mechanism whereby budding yeast cells guarantee that they only establish a single polarity site, growing one and only one bud.

Polarity establishment is thought to occur through a cooperative process involving positive feedback, which allows localized fluctuations in concentration to set off growth of a cluster of polarity factors to establish a front (*Bi and Park, 2012*; *Johnson et al., 2011*). But if stochastic effects can trigger production of a front, what restricts cells to form only one front? A potential mechanism involves competition between different fronts for a common pool of polarity factors (*Goryachev and Pokhilko, 2008*; *Howell et al., 2009*). The strongest experimental support for this competition hypothesis comes from studies of 're-wired' yeast cells that were engineered to use a synthetic polarity factor created from a fusion between two endogenous proteins (*Howell et al., 2009*). In

*For correspondence: daniel. lew@duke.edu

Competing interests: The authors declare that no competing interests exist.

**eLife digest** The cells of fungi and other eukaryotic organisms are generally asymmetric in shape and structure. This is possible because newly-formed cells can establish front and back ends that influence how each cell grows and develops. A protein called Cdc42 – which can cycle between an active and inactive form – establishes this cell "polarity". The front of the cell is specified by the accumulation of active Cdc42 proteins in a small area of the membrane that surrounds the cell.

However, it is not clear why the accumulation of active Cdc42 only happens in one place. Budding yeast is widely used as a model to study cell polarity. This yeast reproduces through the budding of new daughter cells out of one end of the mother cell. To achieve this, the mother establishes a single front end through the accumulation of active Cdc42. This marks the site where the daughter cell will emerge.

Previous research has shown that during the early stages of cell polarization, Cdc42 clusters start to form at several places on the inner surface of the membrane. Only a single cluster emerges as the winner, while the others lose and shrink away. How does this happen? One possibility is that Cdc42 clusters all compete with each other to attract polarity proteins from the cell interior, so that when a protein falls off one cluster it can be picked up by another.

Another possibility is that the winning cluster acquires a factor that stabilizes it, while the other clusters do not. Here, Wu et al. use microscopy and mathematical modelling to distinguish between these two possible scenarios in budding yeast. The experiments show that the speed of competition can be manipulated by altering how quickly key polarity proteins move between the membrane and the interior of the cell. Cells with slowed competition can maintain multiple fronts for long enough to make several buds at the same time.

Wu et al.'s findings suggest that the formation of a single front in budding yeast is due to competition for polarity factors. The next challenge will be to understand how some cells in animals and other fungi manage to suppress this competition in order to produce more than one front.

that system, many cells were observed to initially form two fronts (cortical sites enriched for the synthetic protein). In the majority of cells that developed two fronts, one front then grew stronger while the other concurrently grew weaker and disappeared. When a cell initially developed only one front, that front never shrank or disappeared, suggesting that in the two-front cells, growth of the 'winning' front was responsible for the disappearance of the 'losing' front, as predicted by the competition hypothesis. In a few cells, the two initial polarity sites did eventually grow into buds, indicating that competition is not fully effective in re-wired cells.

Whether competition is responsible for the uniqueness of the front in yeast with a natural (as opposed to synthetic) polarity system is not known. Although we detected initial development of two or more polarity clusters prior to establishment of a single front (*Howell et al., 2012*; *Wu et al., 2013*), others did not (*Klunder et al., 2013*). Moreover, even when a transient multi-cluster intermediate was observed, the process whereby such early intermediates were resolved to a single front remained unclear. Unlike in the strains with a synthetically rewired polarity pathway (*Howell et al., 2009*), in the natural system early polarity clusters were observed to disappear spontaneously even when there was no other cluster present (*Howell et al., 2012*). Thus, the disappearance of a cluster could not be unambiguously attributed to the presence of a competing cluster in the same cell.

Why would some polarity clusters spontaneously disappear? This behavior was traced to a negative feedback loop in the yeast polarity circuit (*Howell et al., 2012*; *Kuo et al., 2014*). As the combination of positive and negative feedback can yield a pulse generator (*Brandman and Meyer, 2008*), it could be that stochastic fluctuations routinely trigger growth of a cluster by positive feedback followed by cluster dissolution due to negative feedback. But if that is the case, then why don't ALL polarity clusters disappear? Why does one and only one cluster remain stable following the initial dynamic behavior? One possibility is that during their brief existence, initial (unstable) polarity clusters have a chance to capture a critical stabilizing factor. Then, once a lucky cluster had captured the stabilizer, all other clusters would be doomed to disappear. Like the competition hypothesis, the stabilizer hypothesis can explain resolution of a multi-cluster intermediate to a final single-front state. Indeed, some models in the field posit that actin cables play roles analogous to the stabilizer,

reinforcing polarity clusters and protecting them from dissolution (*Freisinger et al., 2013*; *Wedlich-Soldner et al., 2003*).

Yeast actin is organized into two distinct types of structures. Actin cables are bundles of parallel actin filaments nucleated by formins: their primary role is to enable myosin-driven delivery of cargo towards the bud (*Pruyne et al., 2004*). Actin patches are assemblies of branched actin filaments nucleated by the Arp2/3 complex: their primary role is to promote internalization of endocytic vesicles (*Kaksonen et al., 2006*). Both actin cable-mediated traffic of secretory vesicles and actin patch-mediated endocytosis have been proposed to stabilize and reinforce polarity clusters (*Freisinger et al., 2013*; *Jose et al., 2013*; *Marco et al., 2007*; *Slaughter et al., 2009*; *Wedlich-Soldner et al., 2004*). When yeast cells were treated with Latrunculin to depolymerize actin, polarity clusters were observed to serially assemble and disassemble, sometimes relocating from one site to another, to a much greater degree than seen in untreated cells (*Howell et al., 2012*; *Okada et al., 2013*; *Wedlich-Soldner et al., 2004*; *Wu et al., 2013*). This observation is consistent with a potential 'stabilizer' role for actin: in cells with two polarity clusters, the first one to capture some actin structure may be stabilized and persist while the other disappears due to negative feedback.

We now report experiments that distinguish between the competition and stabilizer hypotheses. Our findings suggest that uniqueness of the yeast front is due to competition for polarity factors, and not to a downstream stabilizer. We show that the speed of competition can be manipulated by altering the rates at which key polarity factors exchange between membrane and cytoplasm, and that cells with slowed competition can maintain multiple fronts for long enough to make two, three, or even four buds simultaneously. Our findings provide insight into the mechanism of competition, uncovering how yeast cells can guarantee the uniqueness of the front.

## Results

In wild-type yeast cells, polarization is biased towards specific sites by a system of inherited bud-site-selection landmarks (*Bi and Park, 2012*). Localized landmarks influence the site of polarization through the Ras-family GTPase Rsr1, and polarity clusters tend to form near the poles (*Wu et al., 2013*). Because polarity factors also accumulate at the cytokinesis site (which overlaps one pole), some polarity clusters are difficult to quantify separate from the cytokinesis signal. In the absence of Rsr1, polarity clusters can form over much of the cell surface (*Bender and Pringle, 1989*; *Howell et al., 2012*), allowing easier imaging of the resolution from >1 cluster to a single cluster. For that reason, our experiments were carried out in *rsr1* mutant strains.

Because GFP-tagged Cdc42 is not fully functional (*Freisinger et al., 2013*; *Howell et al., 2012*; *Watson et al., 2014*), we adapted a strategy recently shown to produce a functional internal mCherry-tagged Cdc42 in *S. pombe* (*Bendezu et al., 2015*). Although more functional than GFP-Cdc42 at single copy, this probe was still not fully functional in *S. cerevisiae* (*Figure 1A,B*). Thus, when possible we used fluorescently tagged Bem1 as a functional marker for polarity clusters. Bem1 is a scaffold protein that participates in positive feedback (*Kozubowski et al., 2008*) and accumulates at the same sites as Cdc42 with very similar timing (*Howell et al., 2012*); when a losing cluster disassembles, Cdc42 and Bem1 disappear in concert (*Figure 1C*) (*Video 1*).

### Testing candidate stabilizers

According to the stabilizer hypothesis, the difference between a polarity cluster that persists and a cluster that disappears is that the persistent 'winning' cluster acquires a stabilizer, while the disappearing 'losing' cluster does not. Thus, simultaneous imaging of a polarity marker and the stabilizer should reveal the recruitment of the stabilizer to one but not both clusters (*Figure 2A*).

We initially focused on actin cables and actin patches as candidate stabilizers. Actin cables are difficult to visualize directly in live cells (*Huckaba et al., 2004*), so we used two surrogate markers to report cable nucleation and subsequent vesicle delivery by cables. Spa2 is a regulator of the formin Bni1, which nucleates actin cables (*Evangelista et al., 2002*; *Sagot et al., 2002*; *Sheu et al., 1998*); Spa2 recruitment to the polarity site occurs via both actin-dependent and actin-independent routes (*Ayscough et al., 1997*). Sec4 is a secretory vesicle-associated Rab-family GTPase, which polarizes as vesicles are delivered on actin cables to the polarity site (*Mulholland et al., 1997*; *Schott et al., 2002*; *Walch-Solimena et al., 1997*). Spa2-mCherry and GFP-Sec4 both became detectable at the polarity site within about 1 min after Bem1 became detectable (*Figure 2B*). We found that when

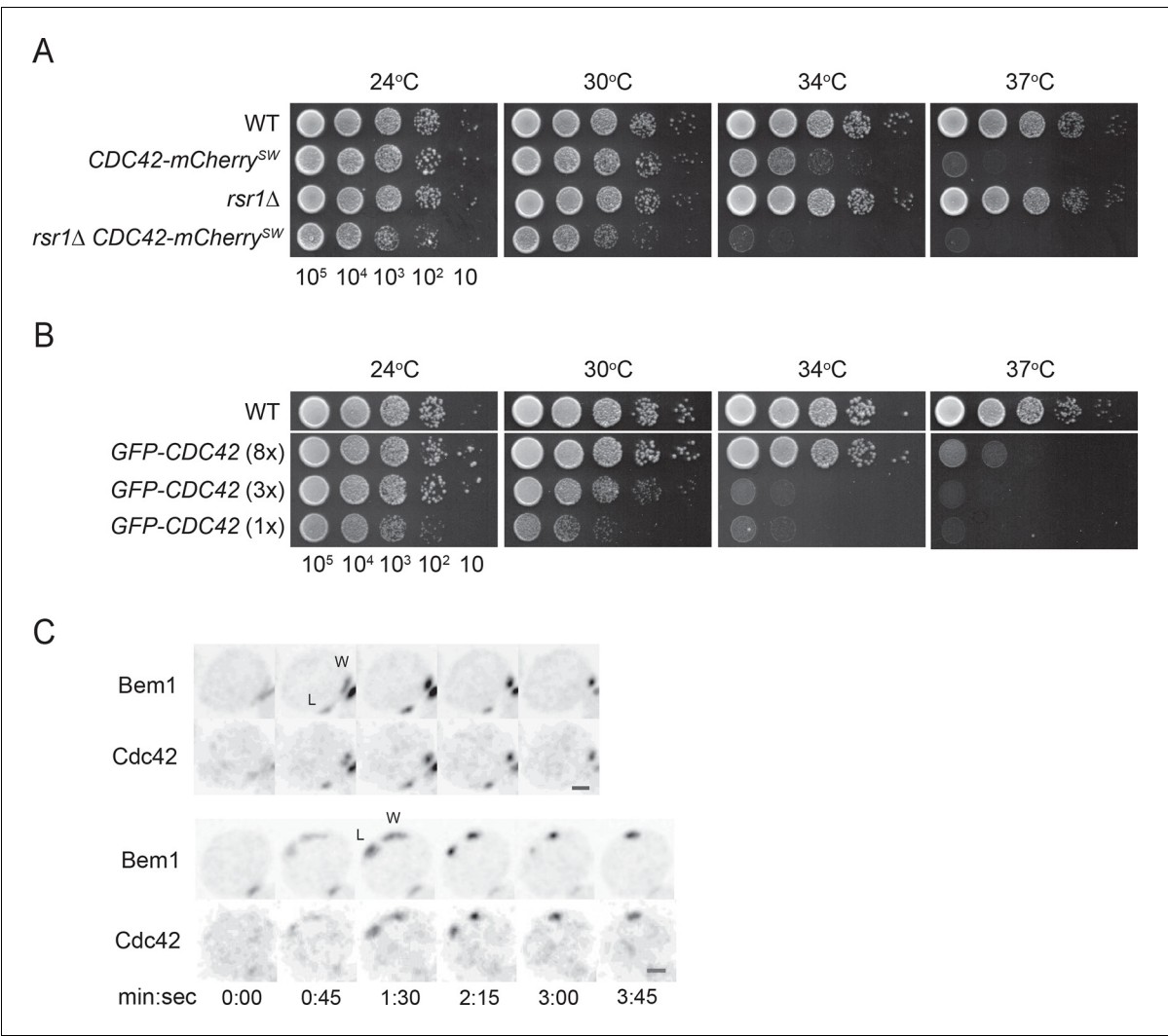

**Figure 1.** Polarity probes. (A,B) Functionality of fluorescent Cdc42 probes. Cells of indicated strains were serially diluted in 10-fold steps from left (10^5 cells) to right, spotted on YEPD plates, and incubated at the indicated temperatures. (A) A construct expressing Cdc42-mCherry^SW from the *CDC42* promoter was integrated at URA3, and the endogenous *CDC42* was deleted. The growth defect of cells expressing only Cdc42-mCherry^SW was more severe in the *rsr1Δ* context. Strains DLY8155, 16855, 5069 and 17127. (B) A construct expressing GFP-Cdc42 is partially functional. Strains carrying GFP-Cdc42 replacing the endogenous Cdc42 showed growth defects at higher temperatures. Higher expression of the probe partially rescued the temperature sensitivity. Strains DLY8155, 13891, 16,730 and 15016. (C) Bem1-GFP and Cdc42-mCherry^SW cluster and disappear concurrently, validating the use of the functional Bem1-GFP as a polarity reporter. Inverted maximum-intensity projections from movies of cells (DLY17110) synchronized by hydroxyurea arrest-release. Time in min:s. L: losing cluster. W: winning cluster.

cells formed two polarity clusters, Spa2 and Sec4 generally accumulated at both sites (*Figure 2C*) (*Video 2*). That is, both the 'winner' (W) and the 'loser' (L) recruited vesicles (and presumably actin cables), indicating that actin cable recruitment does not guarantee persistence of the polarity cluster. Hence, actin cables are unlikely to act as the hypothesized stabilizer.

Actin patches were visualized using the patch marker Abp1 (*Drubin et al., 1988*; *Kaksonen et al., 2003*). Actin patches were initially distributed randomly around the cortex (with some concentration at the old cytokinesis site), and then clustered at the polarity site several minutes after Bem1 became detectable (*Figure 2D*). In most cells that formed two polarity clusters, actin patches remained randomly distributed until one of the clusters had disappeared (*Figure 2E*) (*Video 3*). As neither the winner nor the loser accumulated actin patches during the relevant time-frame, actin patches are also unlikely to act as the stabilizer.

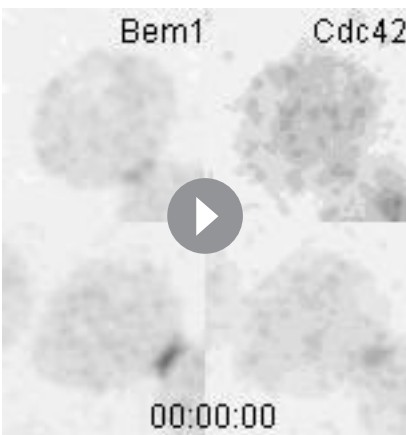

**Video 1.** Rapid resolution of multicluster intermediate during polarity establishment. Strain DLY17110 was imaged following release from HU arrest. Inverted maximum-intensity projections of Bem1-GFP (left) and Cdc42-mCherry$^{SW}$ (right) of two representative cells (upper and lower) are shown. Mother-bud pairs first go through cytokinesis (markers go to the neck), and then polarize both markers to two sites (arrows). One polarity cluster then disappears, leaving a single winner that then fluctuates in intensity and promotes bud emergence. Time in h:min:s.

In addition to actin structures, polarity sites acquire a ring of septin filaments, which then grow to form a very stable hourglass structure at the mother-bud neck (*McMurray and Thorner, 2009*; *Oh and Bi, 2011*). Thus, we considered the possibility that the septin ring might act as a stabilizer. We visualized septin structures using the functional septin probe Cdc3-mCherry (*Caviston et al., 2003*). Septins assembled into a ring around the polarity site several minutes after Bem1 became detectable (*Figure 2F*). In cells that formed two polarity clusters, septins were not readily detectable at either cluster until after one cluster disappeared in most cells (*Figure 2G*). However, we occasionally detected septins at both clusters before one cluster disappeared (*Video 4*). Thus, septins also seem unlikely to act as the stabilizer. Indeed, it has been suggested that septins contribute to negative feedback and cluster destabilization by recruiting Cdc42-directed GAPs (*Okada et al., 2013*).

These findings do not exclude the possibility that some other stabilizer is recruited only to the winning cluster. However, the experiments discussed below allow us to address this possibility more definitively.

## Testing the competition model: reducing polarity protein mobility

If polarity clusters compete with each other for a common pool of polarity factors, then competition would involve transfer of components from the losing cluster to the winning cluster via the cell interior (*Figure 3A*). In this scenario, the relevant factors must exchange dynamically between the cluster and the cell interior on a timescale that is rapid relative to the time it takes to resolve the multi-cluster intermediate. Indeed, fluorescence recovery after photobleaching (FRAP) experiments indicate that polarity factors exchange in and out of clusters on a 2–4 s timeframe (*Freisinger et al., 2013*; *Slaughter et al., 2009*; *Wedlich-Soldner et al., 2004*), whereas multi-cluster resolution occurs on a 1–2 min timeframe (*Howell et al., 2012*). If the exchange of relevant polarity factors in and out of the clusters were to be slowed, then resolution of multicluster intermediates should also occur more slowly. To test this prediction, we generated strains in which Cdc42, or its guanine nucleotide exchange factor (GEF) Cdc24, or the scaffold protein Bem1, exchanged more slowly between membrane and cytoplasm.

Our strategy was based on the expectation that membrane-cytoplasm exchange of a prenyl-anchored protein would be slow compared to that due to the very transient interaction of cytoplasmic proteins with membrane factors. Cdc42 itself is attached to the membrane by a C-terminal polybasic-prenyl motif, but GDP-Cdc42 exchange is rapid due to dedicated factors called Rho guanine nucleotide dissociation inhibitors (Rho-GDIs) (*Johnson et al., 2009*; *Michaelson et al., 2001*). We confirmed previous reports (*Freisinger et al., 2013*; *Slaughter et al., 2009*) that in the absence of the sole yeast Rho-GDI, Rdi1, exchange of Cdc42 in and out of the polarity cluster was much slower (*Figure 3B,C*), while levels of total Cdc42 were similar in wild-type and *rdi1Δ* cells (*Figure 3D*). Biochemical experiments indicated that Cdc42 was able to exchange between different lipid vesicles in vitro even in the absence of a GDI (*Johnson et al., 2009*), and there was still a substantial pool of Cdc42 in the cytoplasm of *rdi1Δ* mutants lacking a GDI, as detected either by fractionation (*Tiedje et al., 2008*) or fluorescence correlation spectroscopy (*Das et al., 2012*). Thus, we anticipated that the slowed Cdc42 dynamics were due to slower exchange of Cdc42 between membrane and cytoplasm, and we fused the Cdc42 polybasic-prenyl motif to the C-termini of Cdc24 and Bem1 (hereafter Cdc24-CAAX and Bem1-CAAX: *Figure 4A*) in order to slow the exchange of these

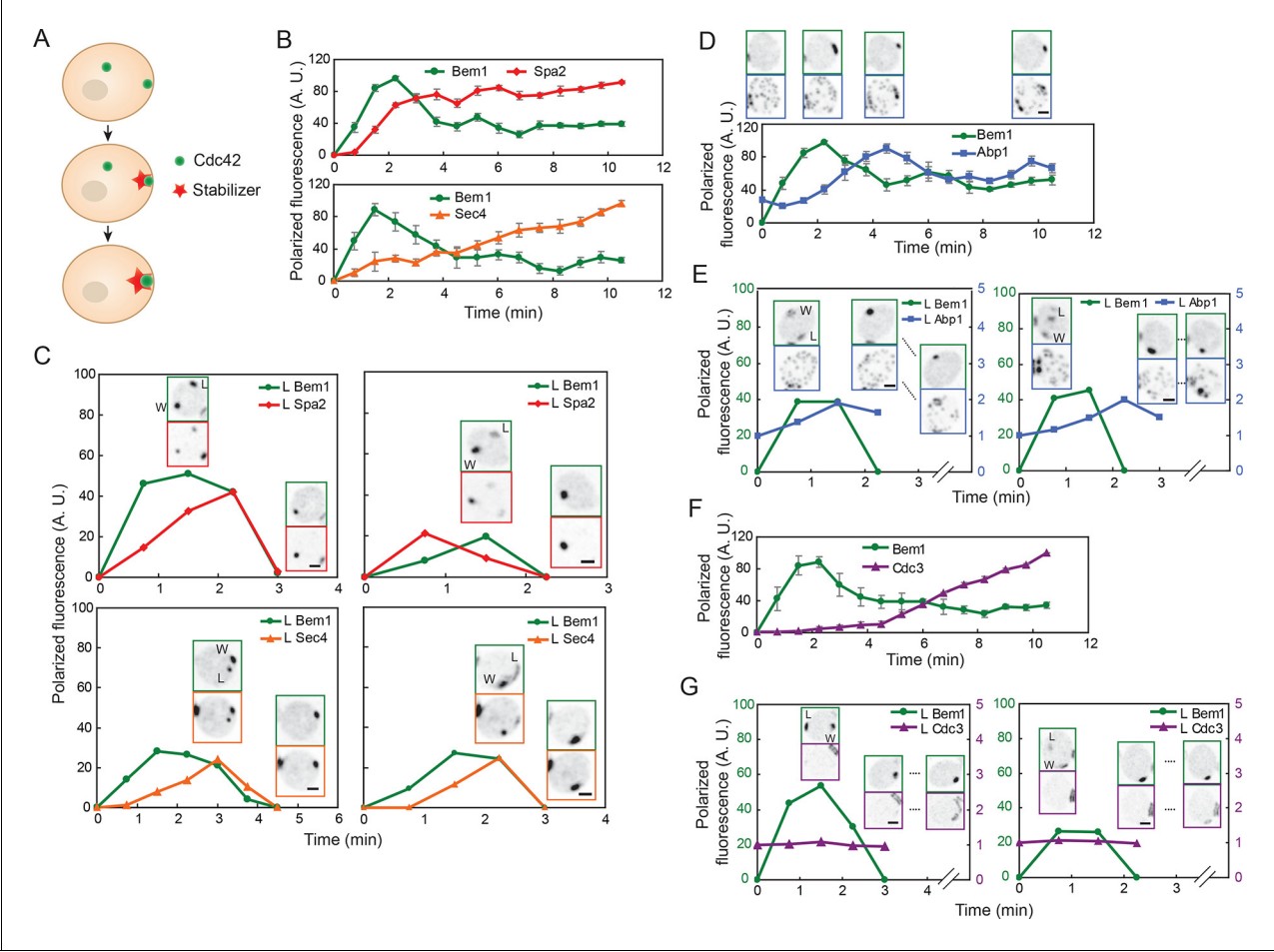

**Figure 2.** Localization of actin cables, actin patches, and septin rings during competition between polarity clusters. (**A**) Stabilizer hypothesis: only the cluster that acquires the stabilizer persists to become the bud site. (**B**) Actin cable markers Spa2-mCherry (upper: DLY17251) and GFP-Sec4 (lower: DLY17374) polarize soon after Bem1-GFP. Data from two-color movies. Summed intensity of the polarized signal is normalized to the peak value within the displayed interval for each cell. t=0 is 45 s before the first detection of polarized signal. Plots show average ± SEM (n=7 cells). (**C**) In cells that have two-cluster intermediate stages, actin cable markers appear at both clusters and then disappear from the losing cluster. Graphs plot summed intensity of Bem1-GFP and Spa2-mCherry (DLY17251) or GFP-Sec4 and Bem1-tdTomato (DLY17374) at the losing cluster, normalized to the peak summed intensity at both clusters. Inset: images of the cells at the indicated times. L: losing cluster. W: winning cluster. (**D**) Clustering of actin patches (marker Abp1-mCherry) at the polarization site is delayed relative to Bem1-GFP. Graph: data from two-color movies (DLY11320) displayed as in (**B**) (n=5 cells). Top: cell snapshots at indicated times from a representative cell. (**E**) In cells that have two-cluster intermediate stages, actin patches do not cluster until after a winner emerges. Graphs plot summed intensity of Bem1-GFP and Abp1-mCherry (DLY11320) at the losing cluster. Inset: images of the cells at the indicated times. L: losing cluster. W: winning cluster. (**F**) Septins (marker Cdc3-mCherry) polarize well after Bem1-GFP. Data from two-color movies (DLY13098) displayed as in (**B**) (n=4 cells). (**G**) In cells that have two-cluster intermediate stages, septins are not recruited until after a winner emerges. Graphs plot summed intensity of Bem1-GFP and Cdc3-mCherry (DLY13098) at the losing cluster. Inset: images of the cells at the indicated times. L: losing cluster. W: winning cluster. Scale bars, 2 μm.

proteins. However, others have argued that in the absence of the GDI, Cdc42 is 'locked on' to cellular membranes, and that the observed exchange of Cdc42 in and out of the polarity site is due to actin-mediated vesicle trafficking (*Freisinger et al., 2013*; *Slaughter et al., 2009*). Thus, we first investigated whether Bem1-CAAX would polarize using membrane-cytoplasm exchange or vesicle trafficking.

In previous work, we fused Bem1 to the exocytic v-SNARE Snc2, a transmembrane protein that becomes polarized by a combination of directed exocytosis, slow diffusion, and efficient endocytosis (*Howell et al., 2009*; *Valdez-Taubas and Pelham, 2003*). This fusion protein was able to replace endogenous Bem1, but created a situation in which formin-nucleated actin cables and actin patch-mediated endocytosis became essential for polarization, because the Bem1-Snc2 protein could only

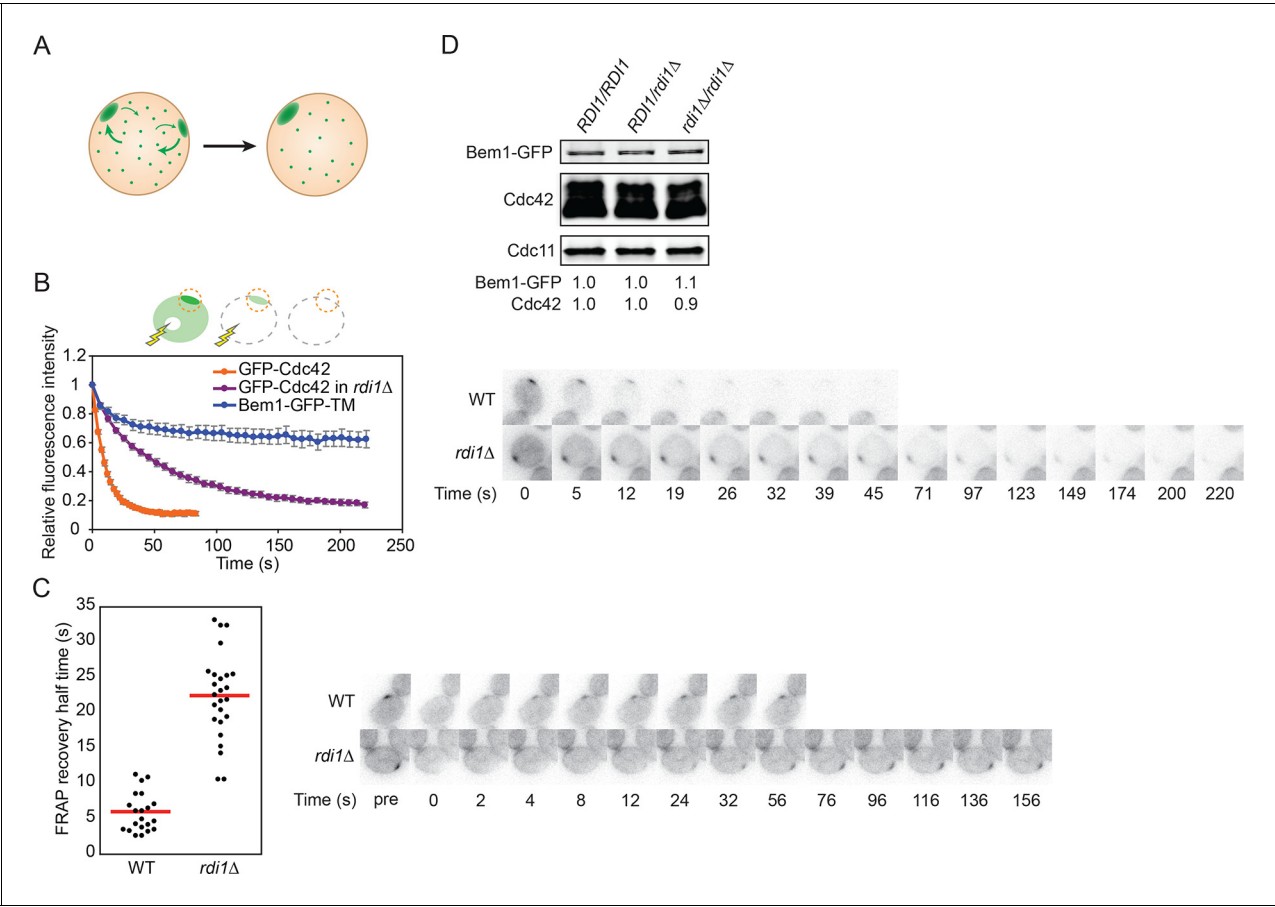

**Figure 3.** Slowing exchange of Cdc42 in and out of polarity clusters. (**A**) Competition hypothesis: clusters compete for shared components from the cell interior. (**B**) FLIP analysis shows that Cdc42 exchanges between membrane and cytoplasm more slowly in *rdi1Δ* cells (DLY14898) than in *RDI1* cells (DLY13920). Bem1-GFP-TM (DLY9641) is a control non-exchanging trans-membrane protein. Cartoon: the laser beam periodically bleached a region of the cytoplasm, and the fluorescence at the polarity patch (dotted red circle) was quantified. Graph: normalized intensity, average ± SEM (n>7 cells). Strips: single z plane snapshots of representative cells at the indicated times. t=0 is right before the first bleaching event. (**C**) FRAP analysis of Cdc42 exchange at the polarized patch in the same cells. The polarized patch was bleached once and the fluorescence recovery measured. Each dot represents the recovery half time of an individual cell. Red lines: average. Strips: single z plane snapshots of representative cells at the indicated times after the initial bleaching. Pre is right before the bleaching event. (**D**) Abundance of Cdc42 and Bem1 are unaffected by the presence or dose of Rdi1. Cdc11 (septin): loading control. Numbers represent Western blot signal normalized to the wild-type. Strains: DLY9200, DLY15241, DLY17301.

traffic on vesicles and not through the cytoplasm (*Chen et al., 2012*; *Howell et al., 2009*). Unlike Bem1-Snc2, however, we found that Bem1-CAAX did not require the formin Bni1 (*Figure 4B,C*) or F-actin (*Figure 4D*) in order to polarize. The finding that Bem1-CAAX polarizes in these situations implies that its mobility is not dependent on actin-mediated vesicle traffic.

In a parallel approach to the same question, we used the 'anchor away' (*Haruki et al., 2008*) system to ask whether Bem1-CAAX was 'locked on' to membranes. This system is based on the ability of the drug rapamycin to induce a stable interaction between FKBP (Fpr1 in yeast) and the FKBP-binding domain (FRB) of Tor1. We fused two tandem copies of FKBP to the ribosomal protein Rpl13A, and two tandem copies of FRB to Bem1-CAAX. Upon addition of rapamycin, this should induce binding of Bem1-CAAX to ribosomes. If Bem1-CAAX is able to exchange between membrane and cytoplasm, then rapamycin should trap it in the cytoplasm, resulting in a loss of Bem1-CAAX from the polarity site. On the other hand, if Bem1-CAAX were locked onto membranes, then rapamycin should not affect Bem1-CAAX localization (though some ribosomes might become attached to the membrane). We found that rapamycin led to a rapid loss of detectable Bem1-CAAX from the polarity site in all cells (*Figure 4E*), providing independent evidence that Bem1-CAAX exchanges between membrane and cytoplasm. In aggregate, these experiments indicate that the

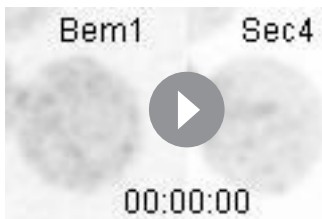

**Video 2.** Vesicle marker Sec4 accumulates at both winning and losing polarity clusters. Strain DLY17374 was imaged following release from HU arrest. Inverted maximum-intensity projections of Bem1-tdTomato (left) and Sec4-GFP (right) are shown. Mother-bud pairs first go through cytokinesis (markers go to the neck), and polarize first Bem1 and then Sec4 to two sites (arrows). One polarity cluster then disappears, leaving a single winner. Time in h:min:s.

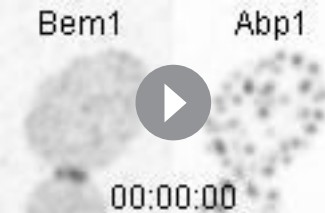

**Video 3.** Actin patch marker Abp1 does not accumulate at polarity clusters until after one cluster has won. Strain DLY11320 was imaged following release from HU arrest. Inverted maximum-intensity projections of Bem1-GFP (left) and Abp1-mCherry (right) are shown. Mother-bud pairs first go through cytokinesis (markers go to the neck), polarize Bem1 to two sites (arrows), and one polarity cluster then disappears, leaving a single winner. Abp1 patches are distributed until one Bem1 cluster wins, after which they accumulate in that vicinity and the bud emerges. Time in h:min:s.

polybasic-prenyl motif slows but does not eliminate membrane-cytoplasm exchange, and that it is valid to use *rdi1Δ* mutants as a way to slow exchange of Cdc42, and Bem1-CAAX and Cdc24-CAAX as a way to slow exchange of Bem1 and Cdc24, between membrane and cytoplasm.

Strains in which Cdc24-CAAX replaced endogenous Cdc24 exhibited very poor viability (*Figure 5A*). Given recent findings that Cdc24 GEF activity can be inhibited by multisite phosphorylation occurring at the membrane (*Kuo et al., 2014*), we wondered whether the Cdc24-CAAX might be nonfunctional due to enhanced inhibitory phosphorylation. Indeed, a mostly nonphosphorylatable Cdc24$^{38A}$-CAAX was viable (*Figure 5A*), although the cells were slower-growing and temperature-sensitive (*Figure 5B*). In contrast, cells in which Bem1-CAAX replaced Bem1 were fully viable and grew well at all temperatures (*Figure 5B*), so in most subsequent experiments we used Bem1-CAAX.

Cdc24$^{38A}$-CAAX and Bem1-CAAX were expressed at comparable levels to Cdc24 and Bem1, respectively (*Figure 5C*). Bem1-CAAX displayed stronger plasma membrane association than Bem1 (*Figure 5D*), and Bem1-CAAX exchange in and out of the polarity site was slower than that of Bem1, as assessed by FLIP or FRAP (*Figure 5E,F*). Bem1-CAAX clusters grew more slowly than Bem1 clusters, and failed to show the characteristic overshoot before reaching their final intensity (*Figure 5G*). This finding suggests that membrane-cytoplasm exchange of Bem1 can (when slowed) become rate-limiting for the growth of polarity clusters. These strains display slowed exchange of key polarity factors between the polarity clusters and the cell interior, allowing us to ask how slowing exchange affects competition between polarity clusters.

## Slowing the exchange of polarity proteins prolongs competition

To test whether slow exchange of polarity factors would delay competition, we conducted time-lapse imaging of the strains discussed above. When two or more polarity clusters formed in any of the slow-exchange strains, the clusters tended to persist for prolonged periods compared to wild-type cells (*Figure 6A–D*). Prolonged coexistence could be documented with any of several

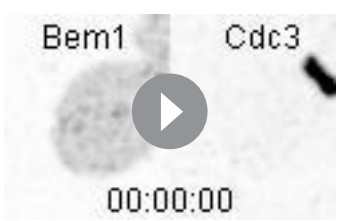

**Video 4.** Septins do not accumulate at polarity clusters until after one cluster has won. Strain DLY13098 was imaged following release from HU arrest. Inverted maximum-intensity projections of Bem1-GFP (left) and Cdc3-mCherry (right) are shown. The septin (Cdc3) starts out at the mother-bud neck, where it is joined by Bem1 as the cell goes through cytokinesis. Bem1 then polarizes to two sites (arrows), and one polarity cluster then disappears, leaving a single winner (a second brief competitor can also be seen at the old neck). After one Bem1 cluster wins (and then fluctuates in intensity), septins accumulate in a ring around the cluster. Time in h:min:s.

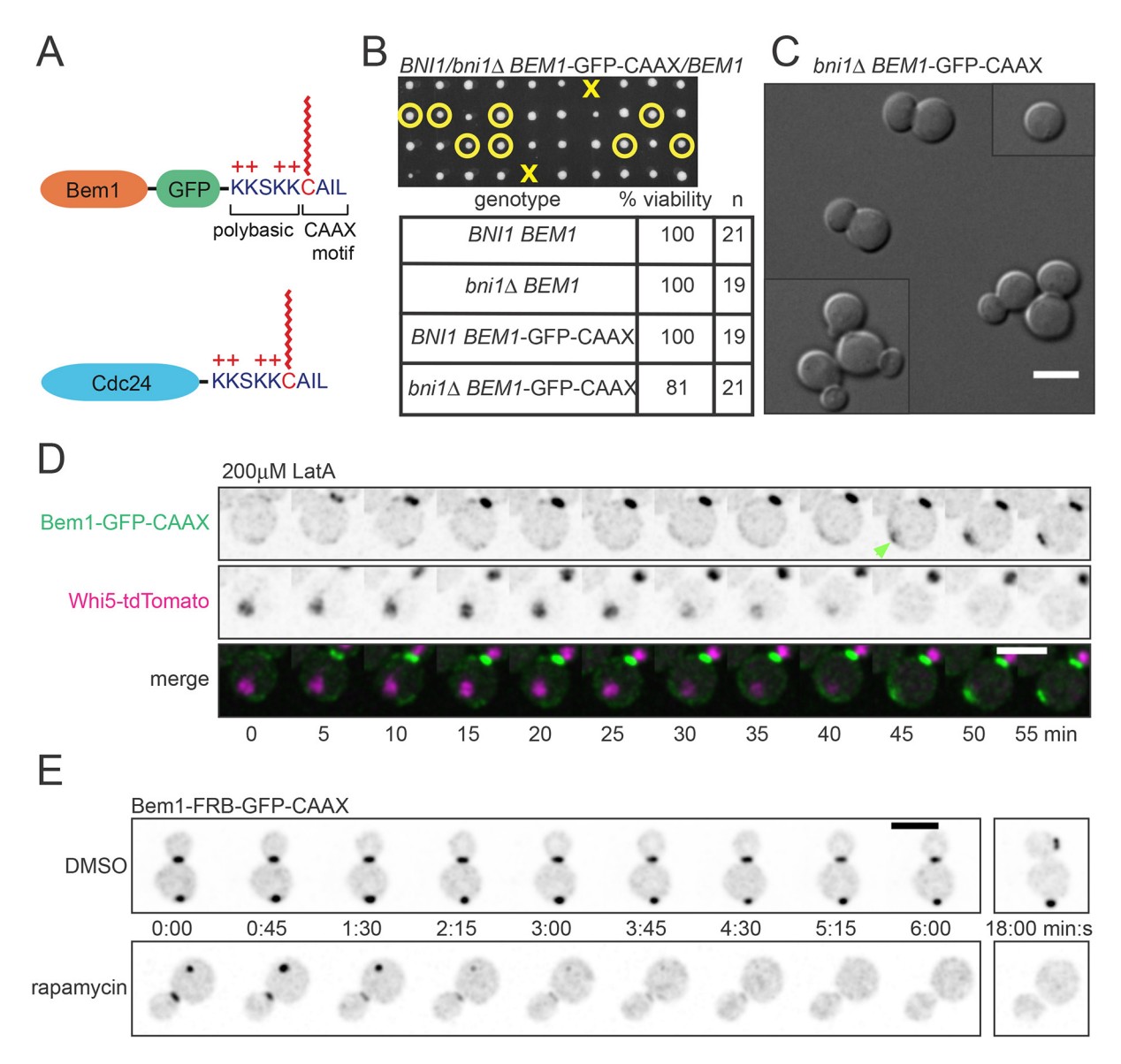

**Figure 4.** The polybasic-prenyl anchor allows slow exchange between membrane and cytoplasm. (A) Strategy: append Cdc42 polybasic-prenyl motif to Bem1 and Cdc24. (B) Cells expressing Bem1-GFP-CAAX as the sole source of Bem1 do not require the formin Bni1. Tetrad dissection from a *BNI1/ bni1Δ BEM1*-GFP-CAAX/*BEM1* diploid (DLY17856). Circles: viable *bni1Δ BEM1*-GFP-CAAX haploids. Crosses: inviable *bni1Δ BEM1*-GFP-CAAX haploids. Table: quantification of% viability. (C) DIC images of viable *bni1Δ BEM1*-GFP-CAAX haploid cells (DLY17859) grown at 24°C. Cells show wide necks typical of *bni1Δ* mutants. Scale bar, 5 μm. (D) Polarization of Bem1-CAAX does not require F-actin. Bem1-GFP-CAAX (top), Whi5-tdTomato (middle), and merged (bottom) images from a representative cell (DLY20283) polarizing in 200 μM LatA at 24°C. The cell-cycle marker Whi5 exits the nucleus upon G1 CDK activation, which provides the signal for polarization (indicated by green arrow). Strips show inverted maximum projections. Scale bar = 5 μm. (E) Bem1-CAAX can be sequestered in the cytoplasm. Rapamycin induces dimerization between FKBP and FRB. Cells containing FKBP-tagged ribosomes and FRB-tagged Bem1-GFP-CAAX (DLY20489) were placed on slabs containing DMSO (top: negative control) or 50 μg/ml rapamycin (bottom) and imaged at 24°C. Binding to cytoplasmic ribosomes delocalizes Bem1-CAAX from polarity sites. Strips show inverted maximum projections. Scale bar, 5 μm.

polarity probes, including GFP-Cdc42, Bem1-GFP, Spa2-mCherry, and PBD-tdTomato (a probe for GTP-Cdc42) (*Tong et al., 2007*) (*Figure 6A–C*). Similar phenotypes were observed for a strain in which Cdc42 was mutated so as to reduce interaction with Rdi1 (*Lin et al., 2003*) (*Figure 6D*). Quantification revealed a heterogeneous range of coexistence times, with average intervals changing

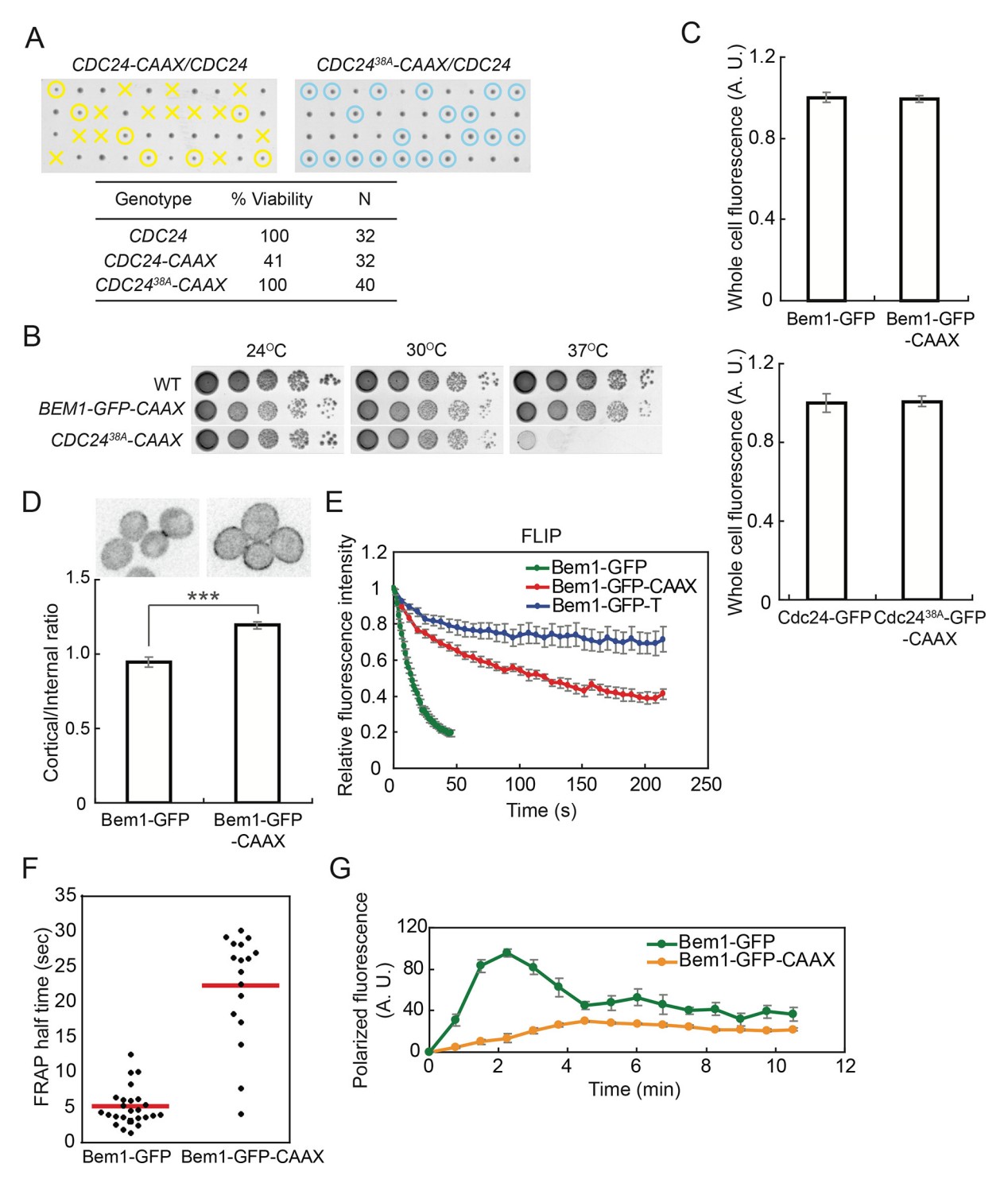

**Figure 5.** Slowing exchange of Bem1 and Cdc24 in and out of polarity clusters. (**A**) Cdc24-CAAX is poorly functional but viability can be rescued by making nonphosphorylatable Cdc24[38A]-CAAX. Tetrad dissection of heterozygotes for *CDC24-CAAX* (DLY18402) or Cdc24[38A]-CAAX (DLY18401): each column has 4 spore colonies from one tetrad. Circles: viable mutants. Crosses: inviable mutants. Table: quantification of% viability. (**B**) Cells with Bem1-CAAX as the sole source of Bem1 (DLY17732) are healthy while those with Cdc24[38A]-CAAX as the sole source of Cdc24 (DLY18565) are temperature-sensitive. (**C**) Appending the polybasic-prenyl motif does not affect abundance of Bem1 or Cdc24. Quantification of whole-cell fluorescence intensity of the indicated GFP-tagged probes (Bem1: DLY11780 and DLY17732; Cdc24:DLY12383 and DLY18417) imaged on the same microscope slab. Average ± SEM of normalized mean intensity per cell (n=11 cells, Bem1; n=14 cells, Cdc24). (**D**) Graph: ratio of cortical to internal fluorescence in strains expressing Bem1-GFP (DLY18920) or Bem1-GFP-CAAX (DLY18849): average ± SEM (n>50 cells). *** p<0.001 by t-test. Top: inverted single-plane
*Figure 5 continued on next page*

*Figure 5 continued*

images of representative cells. (E) FLIP analysis shows that Bem1-GFP-CAAX (DLY17732) exchanges in and out of the polarity site more slowly than Bem1-GFP (DLY9201). Bem1-GFP-TM (DLY9641) is a control non-exchanging trans-membrane protein. Graph: normalized intensity, average ± SEM (n>10 cells). (F) FRAP analysis in the same cells. Each dot represents the recovery half time of an individual cell. Red lines: average. (G) Polarization dynamics: Bem1-GFP-CAAX accumulates more slowly than Bem1-GFP. Summed intensity of the polarized signal is normalized to the peak value within the displayed interval for each cell. Peak levels of polarized Bem1-GFP-CAAX (DLY17732) are lower than those for Bem1-GFP (DLY11780) based on imaging of both strains on same slab, and the graphs were scaled accordingly. t=0 is 45 s before the first detection of polarized signal. Plots show average ± SEM (n=7 cells).

from ~1.5 min in control strains to ~7 min in slow-exchange strains (*Figure 6E*). The coexistence interval could be subdivided into two phases: an initial 'growth'' phase in which two or more clusters all grew in intensity, and a 'competition' phase in which 'losing' clusters shrank and disappeared. Both the growth and competition intervals were longer in slow-exchange strains than in wild-type controls (*Figure 6F*). Thus, slowing the exchange of polarity factors extended the time necessary to resolve multi-cluster intermediates, consistent with a model in which clusters compete for shared components.

## Prolonged competition allows formation of more septin rings and buds

The prolonged competition observed in slow-exchange strains allowed us to ask whether late-arriving factors such as septins are recruited to one or more of the competing clusters. In several cases both winning and losing clusters acquired septin rings (*Figure 7A*) (*Video 5*). However, the presence of septins did not prevent cluster disassembly, and the septin ring also disappeared when a cluster lost the competition (*Figure 7A*). Because we never (n>200) observed disassembly of a septin-containing cluster in cells that did not have another cluster present, it would appear that septin disassembly does not occur spontaneously, and therefore that the disappearance of 'losing' clusters is due to the presence of another cluster, consistent with the competition hypothesis.

In all of the slow-exchange strains, we also encountered cells that formed two buds at the same time (*Figure 7B,C*) (*Video 6*). Simultaneous formation of two buds has been documented previously for *rdi1Δ* mutants, although those investigators had a somewhat different interpretation as to the cause of multibudding (*Freisinger et al., 2013*) (see Discussion). Buds could be similar (*Figure 7B*, cell 1; 7C, cell 1) or dissimilar (*Figure 7B*, cells 2,3; 7C, cell 2) in size, but both buds always emerged at about the same time. This observation indicates that the size difference does not arise because one bud gets a head start; rather, in those cases with different-sized buds competition had proceeded to form unequal clusters at the time of bud emergence, giving one bud a growth advantage. In a few cases, the smaller bud ceased growing (*Figure 7B*, cells 2,3), suggesting that competition continued even after bud emergence, leaving an abandoned bud. We never (n>200) saw a bud stop growing in cells that had only a single bud, suggesting that abandonment of the bud is due to competition with another bud. These findings indicate that the presence of actin and septin structures is unable to stabilize a cluster against competition, arguing strongly against the stabilizer hypothesis.

## Additive effects of combining slow-exchange genotypes

We combined the slow-exchange genotypes discussed above to investigate the effects of simultaneously slowing the exchange of combinations of Cdc42, Cdc24, and Bem1. We were able to combine *rdi1Δ* mutants with either Cdc24[38A]-CAAX or Bem1-CAAX, but combination of Cdc24[38A]-CAAX with Bem1-CAAX proved lethal (*Figure 8A*). *rdi1Δ BEM1-CAAX* strains displayed multibudded cells at increased frequency (*Figure 8B*), as did *rdi1Δ CDC24[38A]-CAAX* strains (though the latter were too sick for accurate quantification). The frequency of multibudded cells in viable strains rose to almost 40% (*Figure 8B*), and some cells grew three or four buds simultaneously (*Figure 8C-E*) (*Video 7*). As discussed above, in a few instances the smallest bud ceased growing, suggesting that competition can continue after bud emergence.

As DNA replication only generates two copies of the genome, cells making more than one bud are unable to pass on a full genetic complement to each daughter. Imaging slow-exchange strains carrying a fluorescent histone revealed that multibudded cells generated anucleate (*Figure 8F*, cell 1) or aneuploid (*Figure 8F*, cells 2 and 3) progeny in which a mother and bud appeared to fight over

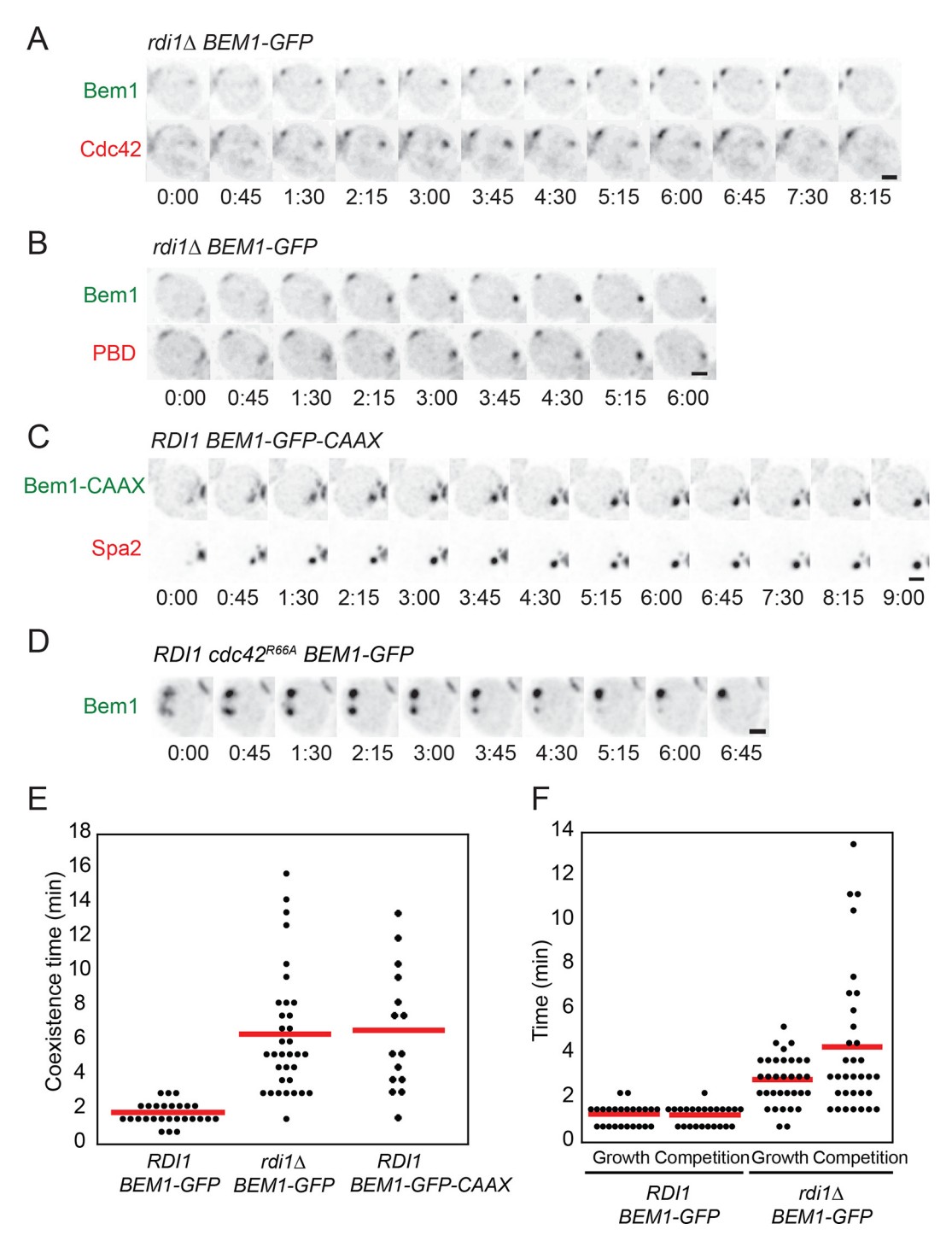

**Figure 6.** Slow competition between polarity clusters in cells with slow membrane/cytoplasm exchange of either Cdc42 or Bem1. Inverted maximum-intensity projections from movies of cells synchronized by hydroxyurea arrest-release. (**A**) Prolonged competition in representative *rdi1Δ* cell (DLY17109) expressing Bem1-GFP and Cdc42-mCherry[SW]. (**B**) Prolonged competition in representative *rdi1Δ* cell (DLY15782) expressing Bem1-GFP and PBD-tdTomato (probe for GTP-Cdc42). (**C**) Prolonged competition in representative *BEM1-GFP-CAAX* cell (DLY12576) expressing Bem1-GFP-CAAX and Spa2-mCherry. (**D**) Prolonged competition in representative *cdc42[R66A]* cell (DLY15572: mutant fails to bind Rdi1) expressing Bem1-GFP. (**E**) Quantification of coexistence intervals (time between the first detection of >1 polarity clusters and disappearance of losing clusters). Each dot represents one cell. Red lines: average. (**F**) Quantification of growth and competition phases. Multiple clusters initially all grow in intensity (growth), after which losing cluster(**s**) shrink and disappear (competition). Time: min:s. Scale bars, 2 μm.

the daughter nuclei (*Video 8*). This observation is rather surprising, and the mechanism by which chromosomes attached to a single spindle pole end up on different sides of the neck remains to be elucidated.

## Mechanism of competition in a computational model

A variety of simple computational models based on biochemical aspects of Rho-family GTPase behavior have illustrated how such GTPases might polarize spontaneously (*Mori et al., 2008*; *Otsuji et al., 2007*; *Semplice et al., 2012*). Like earlier Turing-type models (*Gierer and Meinhardt, 1972*; *Turing, 1952*), some of these can generate and maintain more than one peak of polarity factors in sufficiently large domains. However, a bottom-up model describing the activities and interactions of the yeast Cdc42, Cdc24, Bem1, and GDI proteins displays competition between polarity clusters for all parameters examined thus far (*Goryachev and Pokhilko, 2008*; *Howell et al., 2012*; *Howell et al., 2009*; *Savage et al., 2012*). In this model, whose elements have considerable experimental support (*Kozubowski et al., 2008*), clustering of Cdc42 occurs through a positive feedback loop involving a cytoplasmic complex that contains Bem1 and the GEF Cdc24. Cortical GTP-Cdc42 recruits Bem1-Cdc24 complexes from the cytoplasm, which then load neighboring Cdc42 with GTP, leading to further Bem1-Cdc24 recruitment and Cdc42 activation (*Figure 9A*). Additional Cdc42 is delivered to the growing cluster from the cytoplasm by the GDI, as well as by other pathways (*Johnson et al., 2011*). Because of positive feedback, stochastic activation of a small amount of Cdc42 somewhere on the membrane leads to further accumulation of active Cdc42 until depletion of the cytoplasmic pools of polarity proteins stops the process. With suitable parameter choices, the system develops a stable polarized peak of GTP-Cdc42: diffusion, inactivation, and release of Cdc42 into the cytoplasm is counteracted by recruitment of more Cdc42 to the peak from the cytoplasmic GDI-bound pool (*Figure 9B*). As discussed above, FRAP experiments confirm that apparently stable polarized peaks are indeed maintained by very dynamic recycling of the Cdc42, Bem1, and Cdc24.

The model can be manipulated into generating two peaks if they are initiated with identical stimuli at diametrically opposite poles of the cell. However, this situation is unstable, as the addition of

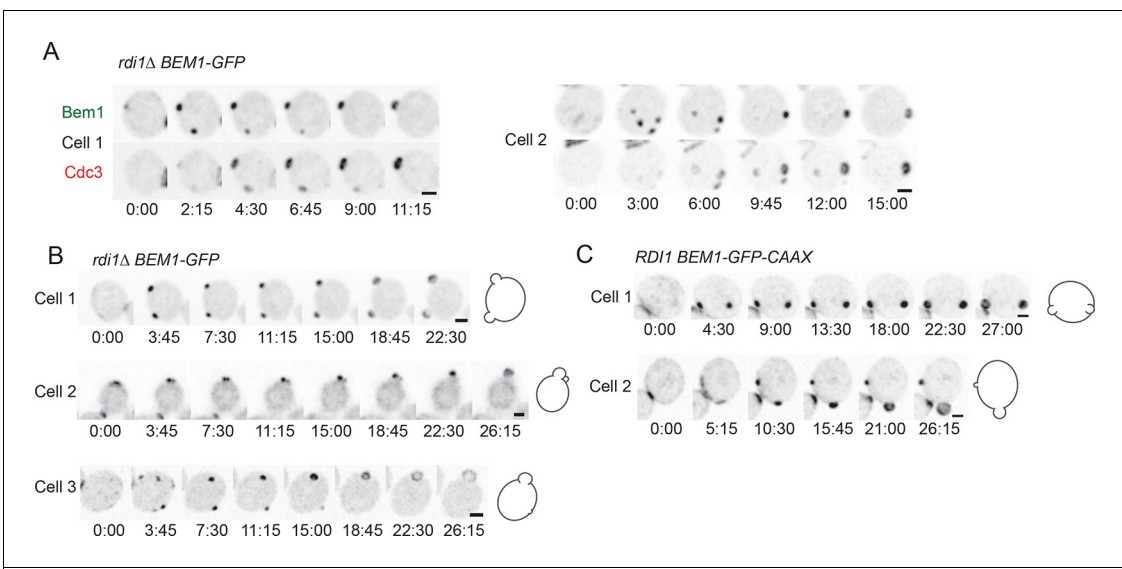

**Figure 7.** Formation of multiple septin rings and buds due to slow competition. (**A**) In cells with slow competition, septins are recruited to multiple polarity clusters but competition continues. Inverted maximum-intensity projections from movies of *rdi1Δ* cells (DLY14535) synchronized by hydroxyurea arrest-release. Representative cells expressing Bem1-GFP and Cdc3-mCherry. (**B**) Simultaneous emergence of two buds in *rdi1Δ* cells (DLY17301) expressing Bem1-GFP. Cell 1: buds far apart, equal size. Cell 2: buds close together. Competition continues after budding (smaller bud abandoned). Cell 3: buds far apart, unequal size. Competition continues after budding (smaller bud abandoned). (**C**) Simultaneous emergence of two buds in *BEM1-GFP-CAAX* cells (DLY17732). Cell 1: buds far apart, equal size. Cell 2: unequal buds, larger grows more rapidly. Cartoons show cell outlines at final timepoint. Scale bars, 2 μm.

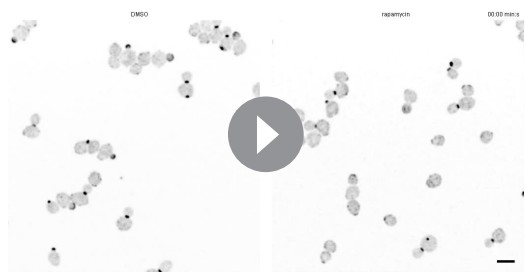

**Video 5.** Sequestering Bem1-CAAX in the cytoplasm Rapamycin induces tight binding between FKBP and FRB. In a strain (DLY20489) where ribosomes are tagged with FKBP (2 copies C-terminal to Rpl13a) and Bem1-GFP-CAAX is tagged with FRB (2 copies between Bem1 and GFP), rapamycin (50 µg/ml, right) delocalized Bem1-GFP-CAAX, but DMSO control (left) did not. Deconvolved, inverted maximum projection images. Time in min:s. Bar, 5 µm.

**Video 6.** Slow resolution of multicluster intermediate in *rdi1Δ* cells allows multiple septin-containing sites to form. Strain DLY14535 was imaged following release from HU arrest. Inverted maximum-intensity projections of Bem1-GFP (left) and Cdc3-mCherry (right) are shown. At least 4 clusters of Bem1 form in this cell, all of which persist long enough to acquire some septins. After a Bem1 cluster disappears, the septins at that site also disappear, leaving a single winner for both Bem1 and Cdc3 (septin). Time in h:min:s.

infinitesimally small noise leads to a stable single-peak steady state (*Figure 9C*) (*Video 9*). At either the two-peak (unstable) or one-peak (stable) steady state, Bem1-Cdc24 complexes and Cdc42 recycle between the peak(s) and the cytoplasm. The net transfer of polarity factors from the 'losing' to the 'winning' peak occurs without significant changes in the cytoplasmic concentrations of Cdc42 and Bem1-GEF for most of the competition time course (*Figure 9D*).

To understand why the two-peak state is unstable, we investigated what happens at the center of each peak when one peak acquires more Cdc42 and Bem1-GEF than the other. We first consider the Bem1-GEF complex. The larger peak has a higher concentration of GTP-Cdc42, which can bind Bem1-GEF from the cytoplasm: this gives it an advantage over the smaller peak in recruiting Bem1-GEF (*Figure 9E*). To evaluate loss of Bem1-GEF from the peak, we started with an arbitrary amount of Cdc42-Bem1-GEF, and ran simulations to monitor the loss of Bem1-GEF from the membrane over time, for different values of GTP-Cdc42 (*Figure 9F*). With higher levels of GTP-Cdc42 (i.e. for larger peaks), it takes longer for Bem1-GEF complexes to detach from the membrane, because when a complex detaches from one molecule of GTP-Cdc42 it is more likely to bind to another GTP-Cdc42 rather than release into the cytoplasm. From these data we extracted the half-life for membrane-bound Bem1-GEF (dwell time), which increased linearly with GTP-Cdc42 (*Figure 9G*). See the Materials and methods for a quasi-steady state approximation demonstrating this effect of GTP-Cdc42 on the Bem1-GEF dwell time.

Now consider the recruitment/removal of Cdc42. To compute the dwell time of Cdc42 as a function of the membrane-bound Bem1-GEF concentration, we used a similar approach as described above for computing the Bem1-GEF loss according to the governing equations (*Figure 9H*, inset). Delivery of Cdc42 from the cytoplasm by the GDI is unaffected by protein concentrations at the membrane, so a similar amount of Cdc42 will be delivered to the center of each peak from the cytoplasm. However, because the larger peak has more Bem1-GEF, GDP-Cdc42 in a larger peak is converted more rapidly to GTP-Cdc42. Because the GDI only extracts GDP-Cdc42, more GEF activity translates to a reduced loss of Cdc42 to the cytoplasm, and hence a longer dwell time (*Figure 9H*). See the Materials and methods for a quasi-steady state

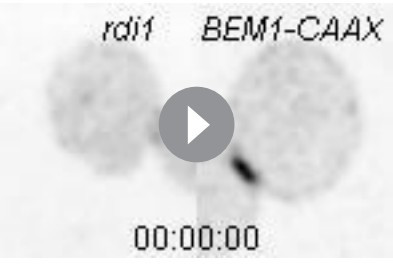

**Video 7.** Cells with slowed exchange of polarity proteins occasionally generate two buds. A representative *rdi1Δ* cell (left, DLY17301, with Bem1-GFP probe) and *BEM1-GFP-CAAX* cell (right, DLY17732) imaged following release from HU arrest. Both cells generated two persistent polarity sites, giving rise equal (left) or unequal (right) buds. Time in h:min:s.

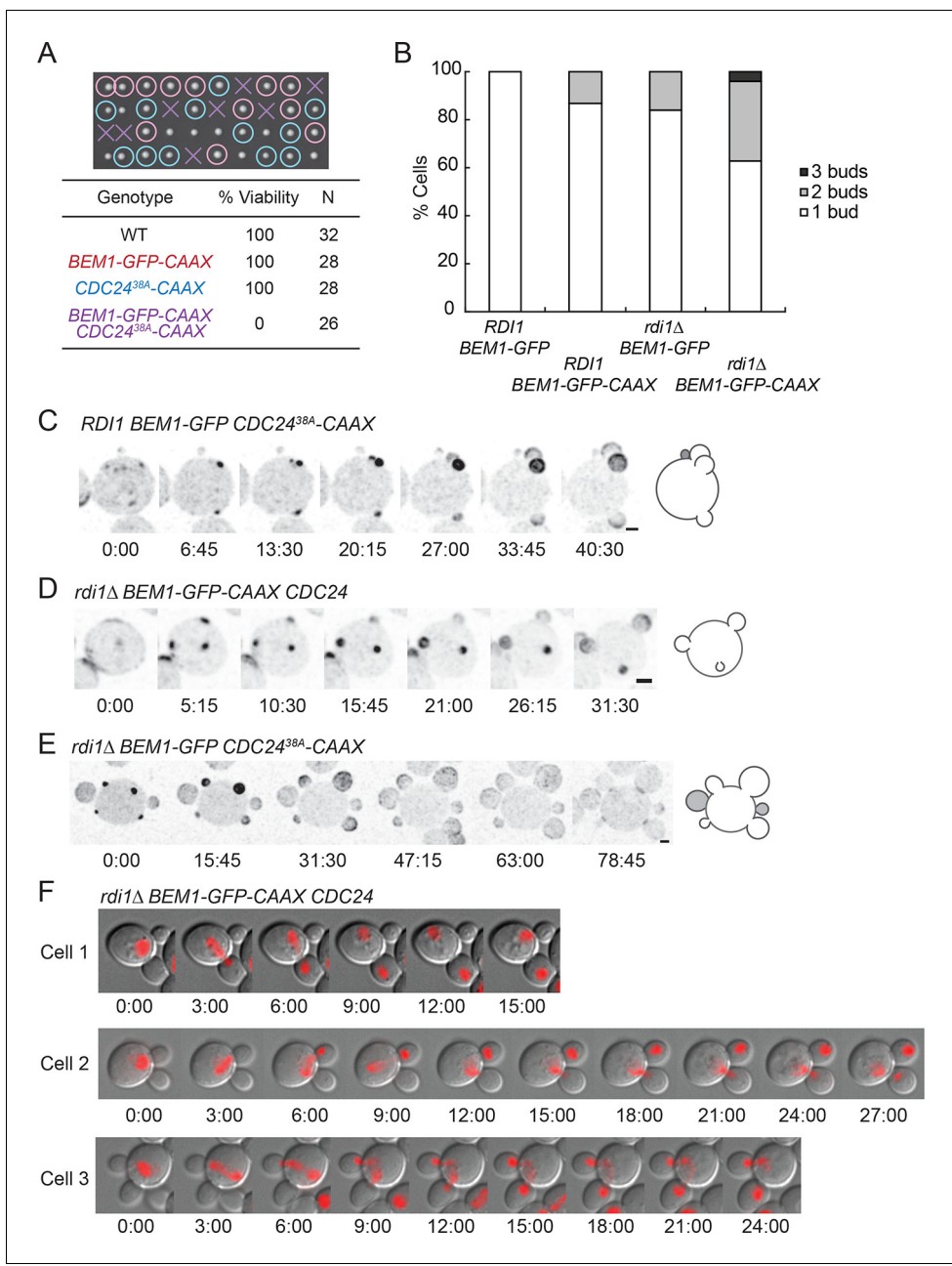

**Figure 8.** Additive effects of combining slow-exchange genotypes. (**A**) Combining *BEM1-GFP-CAAX* and *CDC24³⁸ᴬ-CAAX* is lethal. Tetrad dissection of heterozygotes for *BEM1-GFP-CAAX* and *CDC24³⁸ᴬ-CAAX* (DLY18810): each column has 4 spore colonies from one tetrad. Circles: viable mutants. Crosses: inviable mutants. Table: quantification of% viability. (**B**) Combining *rdi1Δ* with *BEM1-GFP-CAAX* yields increased incidence of multi-budding. Quantification of% cells forming one, two, or three buds simultaneously (n>60 cells for each strain). Strains: DLY17732, DLY17301, DLY17941. (**C**) Simultaneous emergence of three buds in a *CDC24³⁸ᴬ-CAAX* cell (DLY18565) expressing Bem1-GFP. An abandoned bud from the previous cell cycle is indicated in grey. (**D**) Simultaneous emergence of three buds in a *rdi1Δ BEM1-GFP-CAAX* cell (DLY17941). (**E**) Simultaneous emergence of four buds in a *rdi1Δ CDC24³⁸ᴬ-CAAX* cell (DLY18643) expressing Bem1-GFP. Abandoned buds from the previous cell cycle indicated in grey. (**F**) Chromosome segregation in *rdi1Δ BEM1-GFP-CAAX* (DLY18196) cells that make two buds. Chomatin visualized with HTB2-mCherry (histone probe). Cell 1: mother and one bud inherit nuclei, other bud is left vacant. Cells 2 and 3: mothers and buds appear to fight for single nuclei. Scale bars, 2 μm.

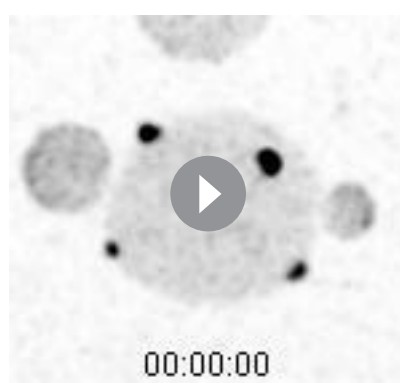

**Video 8.** Simultaneous formation of four buds. An *rdi1Δ CDC24*[38A]*-CAAX* cell expressing Bem1-GFP (DLY18643) was imaged without HU treatment. Four growing buds display concentrated Bem1 while two pre-existing buds on the left and right sides appear to be abandoned buds from the previous cell cycle. Time in h:min:s.

approximation demonstrating how Cdc42 dwell time is related to GEF activity. In summary, the larger peak has an edge in recruiting Bem1-GEF complexes and in retaining both Cdc42 and Bem1-GEF. Thus, the net flux of both species from the cytoplasm to the center of the peak is greater for larger peaks. This mismatch in recruitment and retention for peaks of different sizes provides a mechanism that promotes competition.

Another mechanism that contributes to competition in the model is based on lateral diffusion of polarity factors in the plane of the membrane. As a peak grows or shrinks, its 'waistline' also grows or shrinks in parallel (*Figure 9C*). We define the 'waistline' as the circle at which Cdc42 concentration is half-maximal (i.e. circle diameter is the peak width at half-height) (*Figure 9I*, inset). (The following qualitative argument is not sensitive to the exact definition of the waistline). Monitoring the dissipative flux of Cdc42 due to diffusion across the waistline, we see that a larger peak does not lose as great a proportion of its Cdc42 content as does a smaller peak (*Figure 9I*). Thus, diffusion provides a more powerful dissipative effect for the smaller peak, favoring the larger peak in a competition scenario (*Howell et al., 2009*).

If the diffusional flux of Cdc42 out of the peak is plotted on the same graph as the net recruitment rate of Cdc42 from the cytoplasm into the peak (defined as the area within the waistline), then the intersections of the curves represent steady states, where there is no net change in Cdc42 concentration and the peak size remains constant (*Figure 9J*). From this graph, which is derived from the full simulation of competition in *Video 9*, it is easy to understand why the two-peak solution is unstable. The steady state with two peaks of equal size corresponds to the middle intersection point on the flux plot (*Figure 9J*). If the peaks become slightly unequal, then the diffusional loss is greater than Cdc42 recruitment for the smaller peak (left of intersection point), causing this peak to shrink. However, for the larger peak (right of intersection point), the Cdc42 recruitment flux is greater than the diffusional flux, and this peak grows until the system reaches the one-peak steady state.

In summary, a computational model based on the behavior of the core polarity factors displays competitive behavior because a larger peak has advantages both in terms of reducing diffusional losses and improved recruitment and retention of factors from the cytoplasm. Thus, in a cell with unequal polarity clusters, the largest will grow at the expense of the others.

## Substrate depletion and negative feedback

As polarity factors are recruited to one or more peaks, the cytoplasmic levels of the polarity factors decline, and it is this substrate depletion from the cytoplasm that ultimately stops clusters from growing further. From *Figure 9D*, it is apparent that the cytoplasmic levels of polarity factors at the one-peak steady state are slightly lower than those at the two-peak steady state. Thus, once a single peak has been consolidated, the levels of cytoplasmic factors are too low to support a second peak.

Because substrate depletion is what limits growth in the model, each peak at the two-peak steady state has a lower polarity protein content

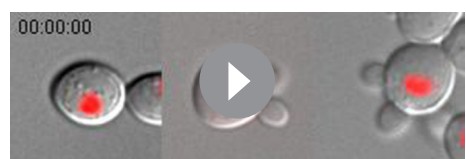

**Video 9.** Chromosome segregation in two-budded cells. An *rdi1Δ Bem1-GFP-CAAX* strain (DLY18196) containing the histone probe HTB2-mCherry to visualize chromatin was imaged following release from HU arrest. Merge of DIC and HTB2-mCherry channels is shown for three representative two-budded cells. Left: chromatin is segregated between the mother and one bud, while the other bud is left vacant. Middle and right: chromatin is split between mothers and buds. Time in h:min:s.

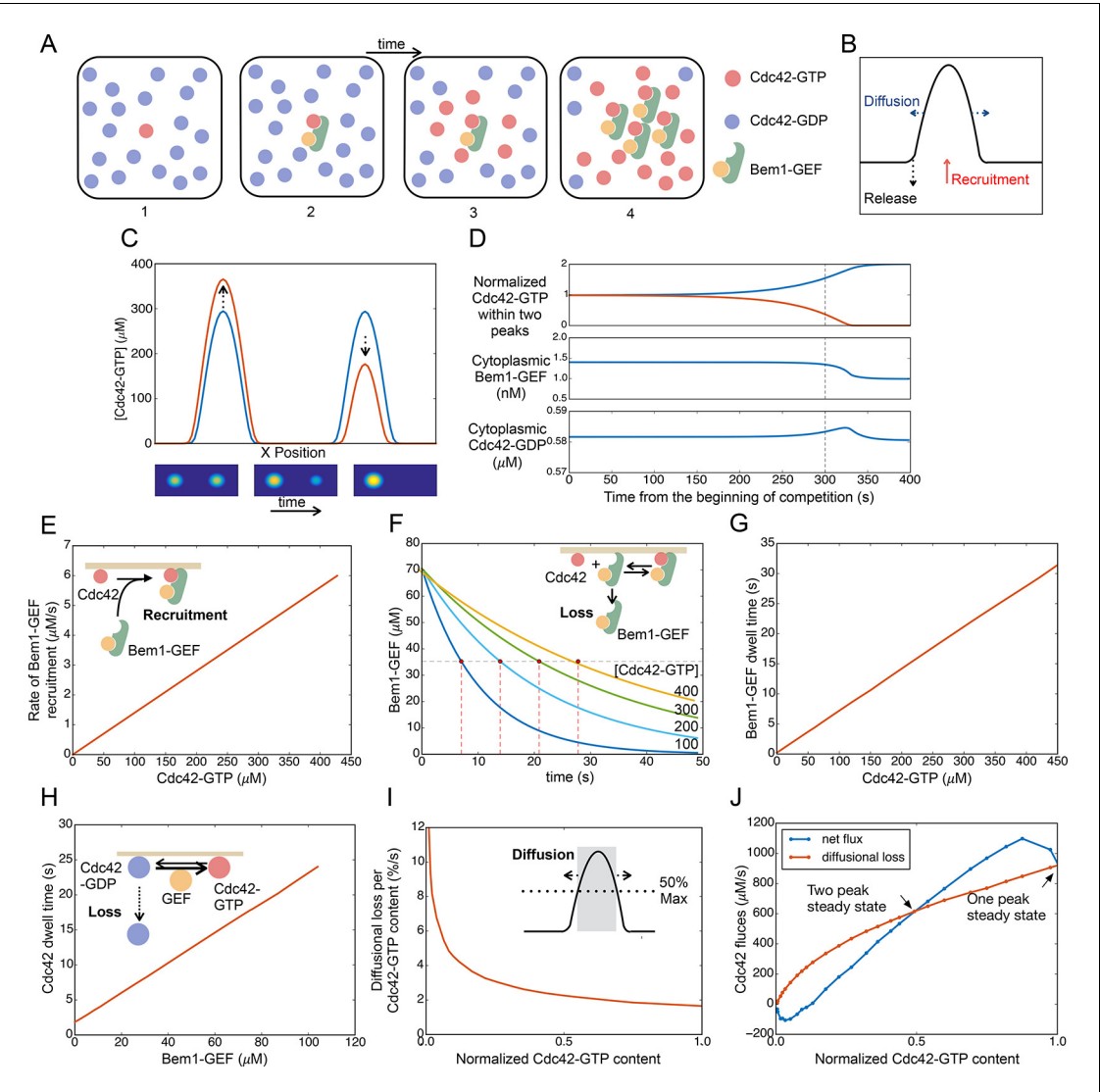

**Figure 9.** Competition between clusters in a computational model. (**A**) Cartoon depicting positive feedback. Snapshots of a patch of plasma membrane in which stochastic activation of Cdc42 (1) leads to binding of Bem1-Cdc24 complex from the cytoplasm (2). Cdc24 (GEF) then loads neighboring Cdc42 with GTP (3), leading to binding of more Bem1-Cdc24 complexes and further Cdc42 activation (4). (**B**) Steady-state polarity peak: polarity protein concentration (Y axis) along the cell perimeter (X axis). The peak is constantly renewed by recruitment of polarity factors from the cytoplasm (red) to combat loss by diffusion (blue) and release of factors back to the cytoplasm (black). (**C**) Simulating competition: two equal peaks (blue) coexist in an unstable steady state: any perturbation drives growth of a winning peak with concomitant shrinkage of the losing peak (red). The graph represents a cross-section of a two-dimensional simulation, for which snapshots are shown below the graph. Color: Cdc42 concentration. (**D**) Top: Starting from an unstable steady state with two equal peaks, one peak (blue) grows larger at the expense of the other (red). During the competition phase (before the dashed line), the cytoplasmic concentrations of both the Bem1-GEF complex (middle) and GDP-Cdc42 (bottom) remained constant. Towards the end, the winning peak grew further and depleted more Bem1-GEF complexes from the cytoplasm. (**E**) Larger peaks have an advantage in recruiting Bem1-Cdc24 complexes. At the center of the peak, the rate of complex recruitment increases with the GTP-Cdc42 concentration. (**F**) Larger peaks have an advantage in retaining Bem1-Cdc24 complexes. Inset: cartoon of the relevant reactions. The curves represent the loss of Bem1-Cdc24 complexes with time, at the indicated GTP-Cdc42 concentrations. Dashed red lines indicate the half-times (dwell times) for each curve. (**G**) The dwell time computed from the simulations in (**F**) increases with the GTP-Cdc42 concentration. (**H**) Larger peaks have an advantage in retaining Cdc42. The dwell time of GDP-Cdc42 was computed for varying GEF concentrations, as described for Bem1-Cdc24. Inset: cartoon of the relevant reactions. (**I**) Larger peaks lose a smaller proportion of their content to lateral diffusion. Rate of escape of Cdc42 from the

*Figure 9 continued on next page*

*Figure 9 continued*

peak by diffusion across the waistline (as a proportion of the Cdc42 content), plotted against the total Cdc42 content within the waistline. Calculated from the simulation in (**C**). Inset: defining a 'waistline' for the polarity peak. (**J**) Rate balance plot for competition between two peaks. The net fluxes of Cdc42 into the peak (recruitment from the cytoplasm: blue) and out of the peak (diffusion: red) from the simulation in (**C**) were plotted against the Cdc42 content within the waistline (normalized to the content of the winning peak). Fluxes are balanced at two steady states: an unstable steady state with two peaks (middle), and a stable steady state with one peak (winner, right; loser, left).

compared to the single peak that emerges from competition. However, in cells this is rarely the case: instead, the winning peak goes on to shed some polarity factors, and in some cases displays oscillations in polarity protein content or even disappears altogether, leading to polarization elsewhere (*Howell et al., 2012*). This behavior has been traced to a negative feedback loop that operates via inhibitory phosphorylation of the GEF Cdc24 (*Kuo et al., 2014*), reducing the level of active GEF available for positive feedback.

Mutants in which the GEF is nonphosphorylatable ($CDC24^{38A}$) largely short-circuit the major negative feedback mechanism, although a slower negative feedback may also occur via Cdc42-directed GAPs (*Kuo et al., 2014*; *Okada et al., 2013*). In $CDC24^{38A}$ mutants, polarity clusters showed competition on similar timescales as that observed in wild-type cells (*Kuo et al., 2014*). Moreover, in $CDC24^{38A}rdi1\Delta$ mutants we observed slow competition and formation of two-budded cells (*Figure 10A*). As predicted by the substrate depletion scenario, cells that made a single bud developed polarity clusters with a higher polarity protein content than those in cells that made two buds (*Figure 10B*). Thus, competition in cells lacking negative feedback proceeds in a manner consistent with the critical features of the model: insatiable positive feedback combined with substrate depletion. The observation that competition proceeds similarly after eliminating a major negative feedback pathway suggests that negative feedback does not greatly affect the competition process in yeast cells.

In computational models that incorporate negative feedback as well as positive feedback, simulations indicate that although competition can proceed in much the same way as discussed above, it is possible to specify parameter values in such a way that the two-peak steady state becomes stable (*Howell et al., 2012*). The basis for this switch in behavior is currently unclear, but likely reflects situations in which the negative feedback loop is sufficiently strong to neutralize the advantage of the larger peak in recruiting polarity factors.

## Emergence of polarity clusters from stochastic fluctuations

The model simulations in *Figure 9* were initiated at a two-peak steady state. If instead simulations are initiated at the homogeneous steady state by addition of random noise, then several small clusters begin to grow and eventually compete, leaving a single winner (*Goryachev and Pokhilko, 2008*). However, a recent modeling study challenged the idea that cluster competition is relevant to yeast polarity establishment, concluding instead that only a single peak of Cdc42, Bem1, and Cdc24 would emerge from initial random noise (*Klunder et al., 2013*). In those simulations, starting noisy distributions of polarity factors merged to form a single very broad but very shallow peak covering an entire hemisphere, which then grew into a single focused peak. The authors used linear stability analysis to demonstrate that only the first mode had a positive

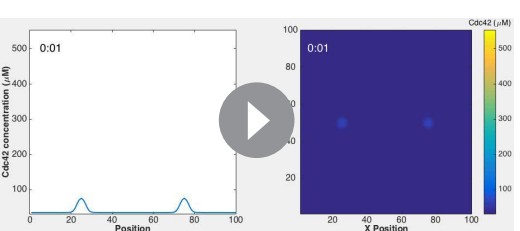

**Video 10.** Simulation of competition between polarity peaks in a computational model. Cross-section (left) and 2D (right: color represents Cdc42 concentration) views of the same simulation. Starting from the homogeneous steady state, two identical perturbations lead to rapid growth of two peaks, which persist for a prolonged period (unstable steady state). Eventually, noise leads to one peak becoming bigger than the other, and this asymmetry leads to accelerating competition until only a single peak persists (stable steady state).

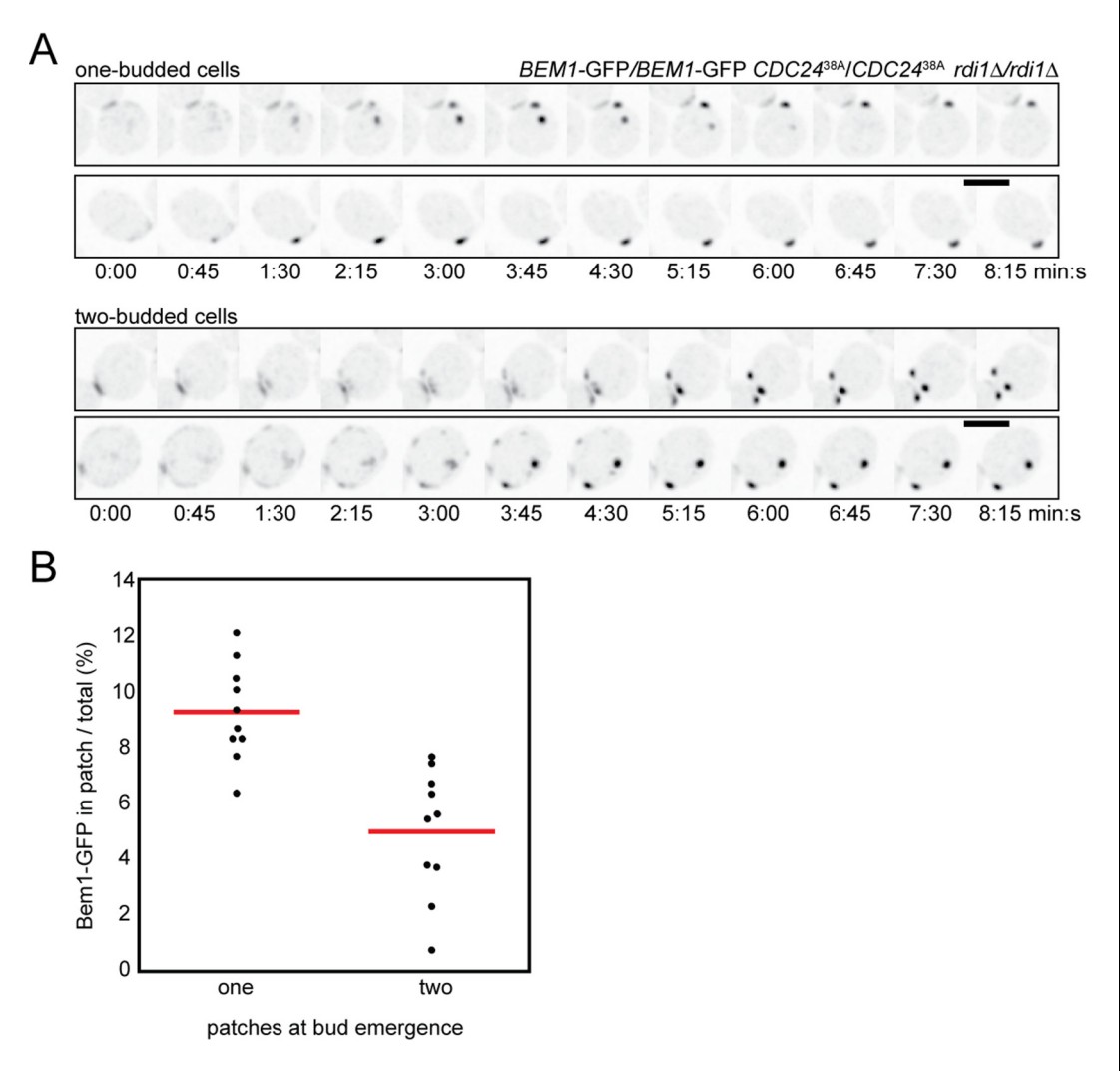

**Figure 10.** Competition in cells with impaired negative feedback. (**A**) Inverted maximum-intensity projections of *CDC24*[38A] *rdi1Δ* cells (DLY18500) expressing Bem1-GFP synchronized by hydroxyurea arrest-release. Top: representative cells that resolve competition and bud once. Bottom: representative two-budded cells. (**B**) Bem1-GFP in the polarity patch immediately before bud emergence was quantitated as a% of the total Bem1-GFP in cells that made one two buds. Each dot represents one patch. Two-budded cells exhibited less Bem1 in each patch compared to one-budded cells.

growth rate, implying that only a single cluster would grow from the homogeneous steady state.

We sought to understand why the different models predicted different behaviors. Although the models are broadly similar and deal with molecular interactions among the same polarity factors, they differ both in the details of how the protein interactions are modeled (*Figure 11A*) and in parameter values (*Figure 11B*). Here we show that the discrepancy stems mainly from how those parameters affect competition versus merging of polarity clusters.

A significant difference between the two models concerns the protein concentrations (*Figure 11B*). In one study (*Klunder et al., 2013*), these were based on estimated molecule numbers per haploid cell as measured by quantitative Western blotting (*Ghaemmaghami et al., 2003*). However, those numbers were applied to a model sphere with volume 258 fL, whereas haploid yeast cells have an average volume of 44 fL (*Klis et al., 2014*). We found that if the molecule numbers were adjusted to account for this volume discrepancy, then higher modes also had a positive growth rate

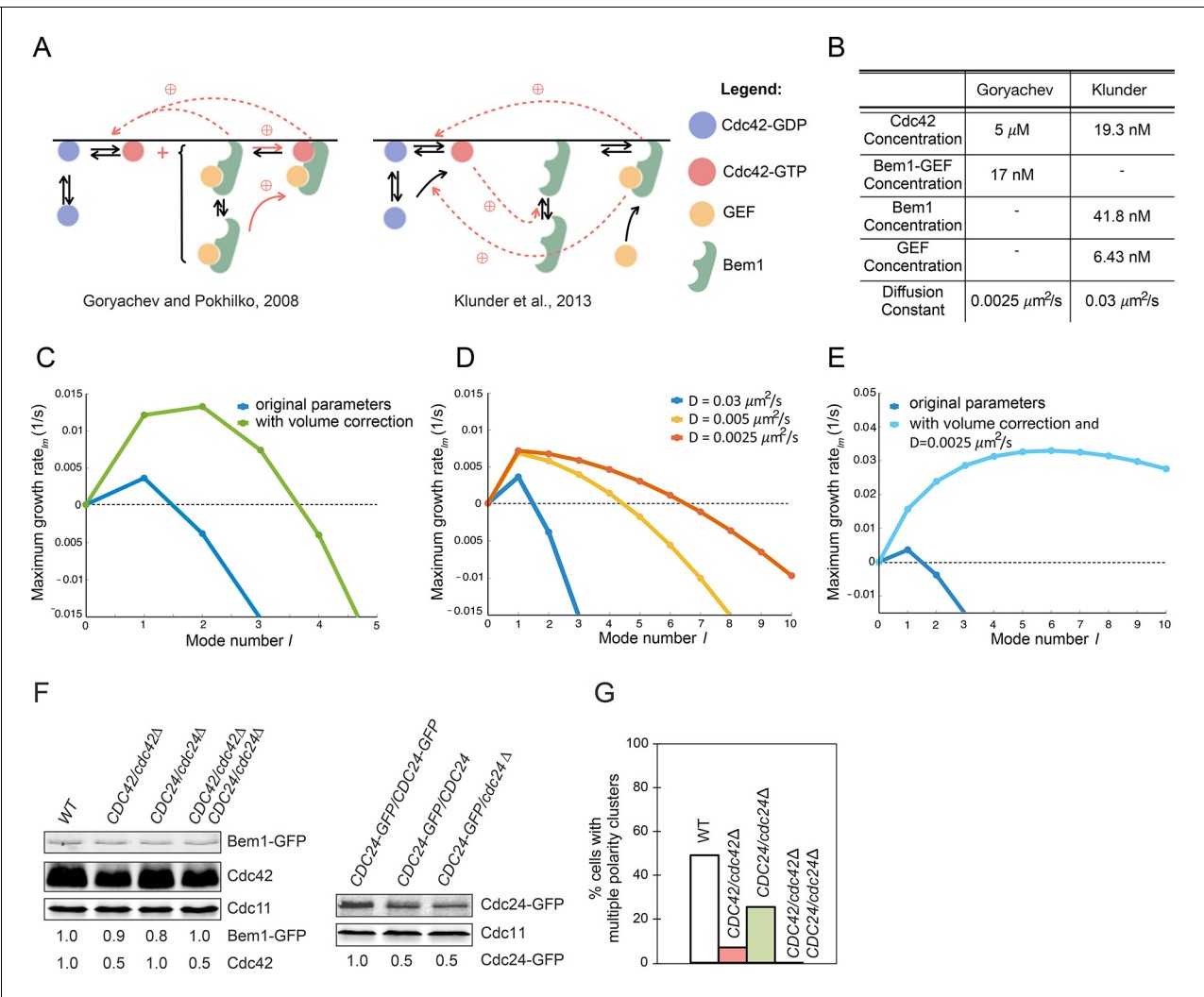

**Figure 11.** Modeling the initial emergence of polarity clusters. (A) Interactions of polarity factors in two published models. (B) Protein concentrations and membrane species diffusion constants in the two models. (C) Increasing protein concentrations would lead to emergence of more than one polarity cluster. Linear stability analysis of the Klunder et al. model. Blue: Klunder et al. parameters. Green: same parameters but correcting the protein concentrations to account for the larger model cell. (D) Effect of slowing diffusion. (E) Effect of increasing protein concentrations as in (C) and slowing diffusion to 0.0025 µm²/s. (F) Reducing gene dosage 2-fold leads to a 2-fold reduction in Cdc42 or Cdc24 levels without affecting Bem1-GFP levels. Western blot analysis of Cdc42, Bem1-GFP, and Cdc24 levels in the indicated strains: DLY9200, DLY13824, DLY17817, DLY18215. Cdc11 (septin): loading control. Numbers represent Western blot signal normalized to the wild-type. (G) Percentage of cells with indicated genotypes (DLY9200, DLY13824, DLY17817, DLY18215) in which a multi-cluster intermediate was detected in movies of cells synchronized by hydroxyurea arrest-release (n>70 cells).

in the linear stability analysis (*Figure 11C*). Thus, with more protein the same model would often yield more than one initial cluster, which would then show competition.

Another difference between the models concerns the estimate of the diffusion constant for membrane-bound species: 0.0025 µm²/s in one study (*Goryachev and Pokhilko, 2008*) and 0.03 µm²/s in the other (*Klunder et al., 2013*). We repeated the linear stability analysis using different values for the diffusion constant, and found that with slower diffusion, higher modes now had a positive growth rate (*Figure 11D*). Combining slow diffusion with higher protein concentrations had a synergistic effect (*Figure 11E*).

These findings demonstrate that the number of clusters likely to emerge from initial noise depends on parameter values. In particular, when polarity concentrations are very low and diffusion is fast in this model, the small initial clusters will tend to merge together before growing to form a

**Table 1.** Yeast strains used in this study.

| Strain | Background | Relevant genotype | Source |
|---|---|---|---|
| DLY5069 | YEF473 | α rsr1::HIS3 | This study |
| DLY8155 | YEF473 | a WT | |
| DLY9200 | YEF473 | a/α rsr1::TRP1/rsr1::TRP1 BEM1-GFP:LEU2/BEM1-GFP:LEU2 | Howell et al., 2009 |
| DLY9201 | YEF473 | a/α BEM1-GFP:LEU2/BEM1-GFP:LEU2 | Wu et al., 2013 |
| DLY9641 | YEF473 | a/α rsr1::HIS3/rsr1::HIS3 BEM1-GFP-snc2$^{V39A,M42A}$:LEU2/BEM1 | Howell et al., 2009 |
| DLY11320 | YEF473 | a/α rsr1::TRP1/rsr1::TRP1 BEM1-GFP:LEU2/BEM1-GFP:LEU2 ABP1-mCherry:kan$^R$/ABP1-mCherry:kan$^R$ | Howell et al., 2009 |
| DLY11780 | YEF473 | a/α rsr1::TRP1/rsr1::TRP1 BEM1-GFP:LEU2/BEM1-GFP:LEU2 SPC42-mCherry:kan$^R$/SPC42 | Howell et al., 2012 |
| DLY12383 | YEF473 | α rsr1::HIS3 CDC24-GFP:TRP1 | This study |
| DLY12576 | | a/α rsr1::HIS3/rsr1::HIS3 BEM1-GFP-CAAX:LEU2/BEM1-GFP-CAAX:LEU2 SPA2-mCherry:kan$^R$/SPA2 | This study |
| DLY13098 | YEF473 | a/α rsr1::TRP1/rsr1::TRP1 BEM1-GFP:LEU2/BEM1-GFP:LEU2 CDC3-mCherry:LEU2/CDC3 | Howell et al., 2012 |
| DLY13824 | YEF473 | a/α rsr1::TRP1/rsr1::TRP1 BEM1-GFP:LEU2/BEM1-GFP:LEU2 cdc42::HIS3/CDC42 | This study |
| DLY13891 | YEF473 | a cdc42::TRP1 URA3:GFP-CDC42 (8x) | This study |
| DLY13920 | YEF473 | a/α rsr1::HIS3/RSR1 cdc42::TRP1/CDC42 URA3:GFP-CDC42/ura3 | This study |
| DLY14535 | YEF473 | a/α rsr1::TRP1/rsr1::TRP1 rdi1::TRP1/rdi1::TRP1 BEM1-GFP:LEU2/ BEM1-GFP:LEU2 CDC3-mCherry:LEU2/CDC3 | This study |
| DLY14898 | YEF473 | a/α rsr1::HIS3/RSR1 rdi1::TRP1/rdi1::TRP1 cdc42::TRP1/CDC42 URA3:GFP-CDC42/ura3 | This study |
| DLY15016 | YEF473 | a GFP-CDC42 | This study |
| DLY15121 | YEF473 | a/α rdi1::TRP1/rdi1::TRP1 BEM1-GFP:LEU2/BEM1-GFP:LEU2 | This study |
| DLY15241 | YEF473 | a/α rsr1::HIS3/rsr1::HIS3 rdi1::TRP1/RDI1 BEM1-GFP:LEU2/BEM1-GFP:LEU2 | This study |
| DLY15782 | YEF473 | a/α rsr1::HIS3/rsr1::HIS3 rdi1::TRP1/rdi1::TRP1 BEM1-GFP:LEU2/ BEM1-GFP:LEU2 PBD-tdTomato:kan$^R$/GIC2 | This study |
| DLY15572 | YEF473 | a/α rsr1::TRP1/rsr1::TRP1 BEM1-GFP:LEU2/BEM1-GFP:LEU2 cdc42$^{R66A}$/cdc42$^{R66A}$ | This study |
| DLY16730 | YEF473 | α cdc42::TRP1 URA3:GFP-CDC42 (3x) | This study |
| DLY16855 | YEF473 | a cdc42::TRP1 URA3:CDC42-mCherry$^{SW}$ | This study |
| DLY17109 | YEF473 | a/α rsr1::HIS3/rsr1::HIS3 rdi1::TRP1/rdi1::TRP1 BEM1-GFP:LEU2/ BEM1-GFP:LEU2 cdc42::TRP1/CDC42 URA3:CDC42-mCherry$^{SW}$/ura3 | This study |
| DLY17110 | YEF473 | a/α rsr1::HIS3/rsr1::HIS3 BEM1-GFP:LEU2/BEM1-GFP:LEU2 cdc42::TRP1/CDC42 URA3:CDC42-mCherry$^{SW}$/ura3 | This study |
| DLY17127 | YEF473 | α rsr1::HIS3 cdc42::TRP1 URA3:CDC42-mCherry$^{SW}$ | This study |
| DLY17251 | YEF473 | a/α rsr1::TRP1/rsr1::TRP1 BEM1-GFP:LEU2/BEM1-GFP:LEU2 SPA2-mCherry:kan$^R$/SPA2 | This study |
| DLY17301 | YEF473 | a/α rsr1::HIS3/rsr1::HIS3 rdi1::TRP1/rdi1::TRP1 BEM1-GFP:LEU2/ BEM1-GFP:LEU2 | This study |
| DLY17374 | YEF473 | a/α rsr1::HIS3/rsr1::HIS3 BEM1-tdTomato:HIS3/BEM1 GFP- URA3:SEC4/ura3 | This study |
| DLY17675 | YEF473 | a/α rsr1::HIS3/rsr1::HIS3 rdi1::TRP1/rdi1::TRP1 cdc42::TRP1/CDC42 URA3:GFP-CDC42/ura3 | This study |
| DLY17732 | YEF473 | a/α rsr1::HIS3/rsr1::HIS3 BEM1-GFP-CAAX:LEU2/BEM1-GFP-CAAX:LEU2 | This study |
| DLY17817 | YEF473 | a/α rsr1::TRP1/rsr1::TRP1 BEM1-GFP:LEU2/BEM1-GFP:LEU2 cdc24::URA3/CDC24 | This study |

*Table 1 continued on next page*

*Table 1 continued*

| Strain | Background | Relevant genotype | Source |
|---|---|---|---|
| DLY17856 | BF264-15Du | a/α bni1::URA3/BNI1 rsr1::kan$^R$/RSR1 BEM1-GFP-CAAX:LEU2/BEM1 bar1/BAR1 | This study |
| DLY17879 | BF264-15Du | a bni1::URA3 rsr1::kan$^R$ BEM1-GFP-CAAX:LEU2 | This study |
| DLY17941 | YEF473 | a/α rsr1::HIS3/rsr1::HIS3 rdi1::TRP1/rdi1::TRP1 BEM1-GFP-CAAX:LEU2/ BEM1-GFP-CAAX:LEU2 | This study |
| DLY18196 | YEF473 | a/α rsr1::HIS3/rsr1::HIS3 rdi1::TRP1/rdi1::TRP1 BEM1-GFP-CAAX:LEU2/ BEM1-GFP-CAAX:LEU2 HTB2-mCherry:nat$^R$/HTB2 | This study |
| DLY18215 | YEF473 | a/α rsr1::TRP1/ rsr1::TRP1 BEM1-GFP:LEU2/ BEM1-GFP:LEU2 cdc42:: HIS3/CDC42 cdc24::URA3/CDC24 | This study |
| DLY18401 | YEF473 | a/α rsr1::TRP1/rsr1::TRP1 CDC24$^{38A}$-CAAX:kan$^R$/CDC24$^{38A}$ | This study |
| DLY18402 | YEF473 | a/α rsr1::TRP1/rsr1::TRP1 CDC24-CAAX:kan$^R$/CDC24 | This study |
| DLY18417 | YEF473 | α rsr1::TRP1 CDC24$^{38A}$-CAAX:kan$^R$ | This study |
| DLY18565 | YEF473 | a/α rsr1::TRP1/rsr1::TRP1 BEM1-GFP:LEU2/BEM1-GFP:LEU2 CDC24$^{38A}$-CAAX:kan$^R$/CDC24$^{38A}$-CAAX:kan$^R$ | This study |
| DLY18643 | YEF473 | a/α rsr1::TRP1/rsr1::TRP1 rdi1::TRP1/rdi1::TRP1 BEM1-GFP:LEU2/ BEM1-GFP:LEU2 CDC24$^{38A}$-CAAX:kan$^R$/CDC24$^{38A}$-CAAX:kan$^R$ | This study |
| DLY18649 | YEF473 | a/α HTB2-mCherry:nat$^R$/HTB2 rsr1::TRP1/RSR1 CDC24$^{38A}$-GFP-CAAX: nat$^R$/CDC24 | This study |
| DLY18663 | YEF473 | a HTB2-mCherry:nat$^R$ CDC24$^{38A}$-GFP-CAAX:nat$^R$ | This study |
| DLY18810 | YEF473 | a/α BEM1-GFP-CAAX:LEU2/BEM1 CDC24$^{38A}$-CAAX:kan$^R$/CDC24 | This study |
| DLY18849 | YEF473 | a/α rsr1::HIS3/rsr1::HIS3 BEM1-GFP-CAAX:LEU2/BEM1-GFP-CAAX:LEU2 LEU2:pTEF1-PRS1(1-208)-mCherry/leu2 | This study |
| DLY18859 | YEF473 | a/α rsr1::HIS3/RSR1 cdc42::TRP1/CDC42 URA3:GFP-CDC42/ura3 | This study |
| DLY18920 | YEF473 | a/α rsr1::TRP1/rsr1::TRP1 BEM1-GFP:LEU2/BEM1-GFP:LEU2 LEU2: pTEF1-PRS1(1-208)-mCherry/leu2 | This study |
| DLY20383 | YEF473 | a rsr1::HIS3 BEM1-GFP-CAAX:LEU2 WHI5-mCherry::URA3 | This study |
| DLY20489 | YEF473 | a rsr1::TRP1 BEM1-2xFRB-HA-GFP-CAAX:LEU2:nat$^R$ fpr1::kan$^R$ tor1-1 RPL13a-2xFKBP-HA | This study |

single detectable peak. Slower diffusion, or the more powerful positive feedback that occurs in the model when polarity factors are more abundant, can lead to growth of separate clusters before they have a chance to merge. Resolution of those clusters then occurs by competition in both models.

A prediction of these computational findings is that if polarity factor concentrations were lowered, then multi-cluster intermediates should be less prevalent. To test this prediction, we imaged diploid strains in which one copy of *CDC42* or *CDC24* was deleted. Western blotting showed that hemizygotes contained half as much Cdc42 or Cdc24 as homozygotes (*Figure 11F*). We monitored polarity establishment in these strains using a Bem1-GFP probe, whose abundance was similar in all strains (*Figure 11F*). Whereas we detected more than one initial cluster in about 50% of wild-type cells, multi-cluster intermediates were detected in only 25% of *CDC24* hemizygotes and 5% of *CDC42* hemizygotes (n>100 cells for each strain) (*Figure 11G*). No multi-cluster instances were observed in cells doubly hemizygous for both *CDC42* and *CDC24* (n=73) (*Figure 11G*). In separate experiments, we detected 30% fewer instances of multicluster intermediates in *BEM1-GFP/bem1Δ* hemizygotes than in *BEM1-GFP/BEM1-GFP* homozygotes. Thus, multiple clusters are less frequent in cells that express lower levels of polarity factors.

## Discussion

Most polarized cells generate only one front. Our findings indicate that in yeast, this rule is enforced by a greedy competition between potential polarity sites to accumulate polarity factors.

We detected multiple polarity clusters as an intermediate stage in polarity establishment in approximately 50% of cells under our imaging conditions (see also *Howell et al., 2012*). As clusters

can occur anywhere on the cell surface and resolution to a single cluster is typically rapid, the frequency with which we detect such intermediates will clearly depend on the spatiotemporal resolution at which imaging is conducted. This may explain why another recent study detected many fewer multi-cluster intermediates when imaging only the medial planes of large cells (*Klunder et al., 2013*). In addition, there may be strain background differences in multi-cluster frequency, as we found that two-fold reductions in polarity factor abundance reduced the incidence of multi-cluster intermediates considerably.

Why would polarity protein abundance be correlated with the incidence of multi-cluster intermediates? When polarity factor concentration is low, small initial clusters grow more slowly, perhaps allowing more time for diffusion-based merging of nearby clusters to form a broad and shallow single peak (*Klunder et al., 2013*). However, it is unclear whether merging is sufficient to explain the reduced incidence of multi-cluster intermediates. In our slow-exchange mutant strains, clusters often co-existed in close proximity for prolonged periods. This suggests that merging is inefficient, presumably because diffusion is very slow in the yeast plasma membrane (*Valdez-Taubas and Pelham, 2003*). An alternative hypothesis is that whereas the models display hair-trigger Turing instability, in the cells it may take more than just a tiny asymmetry to set off growth of a cluster. Indeed, inclusion of negative feedback can produce this effect in the model (*Howell et al., 2012*). If stochastic events need to cross some threshold of local polarity factor concentration in order to trigger growth of a cluster, then the frequency of such stochastic events may be quite sensitive to polarity protein concentration.

Given that cells frequently develop more than one initial cluster of polarity factors, there must be a mechanism to eliminate excess clusters so that only one persists to form the front. We suggest that this mechanism involves competition between polarity clusters for components including Cdc42, Bem1, and Cdc24. Each of these factors exchanges constantly and rapidly (2–5 s half-time) between the polarity cluster and the cell interior (*Freisinger et al., 2013*; *Slaughter et al., 2009*; *Wedlich-Soldner et al., 2004*). Polarity factors released from one cluster may be captured by another, and if larger clusters have an advantage in recruiting and retaining such factors, then they would grow at the expense of smaller clusters. Consistent with that hypothesis, we found that slowing the exchange of Cdc42, Bem1, or Cdc24 in and out of the clusters resulted in correspondingly slower resolution of multi-cluster intermediates, leading to the occasional formation of more than one bud. Combinatorial slowing of polarity factor exchange had additive effects, yielding strains that frequently made more than one bud.

The finding that *rdi1Δ* mutants occasionally make two buds was also reported recently by another group (*Freisinger et al., 2013*). Those authors suggested that in the absence of *RDI1*, polarity establishment occurs through a pathway involving actin cables, and that once actin cables attach at a particular site the cell is committed to making a bud there. Our data argue against this hypothesis: we found that several polarity clusters could recruit actin cables (as judged by delivery of vesicles) in both wild-type and *rdi1Δ* mutant cells, but this did not prevent elimination of most clusters. Slow resolution of multi-cluster intermediates also continued in *rdi1Δ* and other slow-exchange strains after septins had been recruited, and even in some instances after buds had begun to grow. Thus, it would seem that neither actin cables nor indeed any other 'stabilizer' acts to lock in a polarity site in cells that have other polarity sites. Notably, slow-exchange mutant cells that developed only one cluster never eliminated that cluster. These findings imply that it is the presence of a competing cluster that promotes the dissolution of other clusters in the same cell.

In principle, clusters could actively inhibit other clusters in the same cell, rather than simply competing for shared components. Indeed, this type of mechanism has been proposed to explain why neutrophils maintain only one front (*Houk et al., 2012*). As in yeast, more than one front can transiently co-exist in neutrophils. Each front promotes actin polymerization and membrane protrusion, leading to increased membrane tension, which in turn appears to inhibit GTPase activation. Thus, tension promoted by a dominant GTPase cluster actively extinguishes smaller clusters. This seems unlikely to account for the elimination of excess Cdc42 clusters in yeast. First, membrane tension in yeast (and other walled cells) is determined by turgor pressure rather than actin polymerization. Second, if a yet to be identified inhibition mechanism was functioning in yeast, it is not obvious why slowing the exchange of polarity factors would counteract it. Thus, the simplest hypothesis is that elimination of excess clusters reflects the depletion of polarity factors from losing clusters as they are acquired by a competing cluster.

As the slow-exchange mutant cells could make up to four buds simultaneously without overexpression or any increase in ploidy, all cell components required for budding must be present in considerable excess of what is required to make a functioning polarity site. If there are sufficient polarity factors to make several functional fronts, why is it that in wild-type cells, competition continues until there is only a single winner? Analyses of a computational model incorporating some of the known interactions among yeast polarity factors suggests that a larger cluster would have significant advantages over a smaller cluster in both recruiting and retaining polarity factors. The insatiably acquisitive nature of this competitive process would lead to an inexorable rich-get-richer spiral in which the winning cluster starves all others of polarity factors. This behavior has clear parallels (though with differences in mechanism) to coarsening phenomena in physics (*Semplice et al., 2012*).

Is competition for polarity factors also relevant to other situations in which cells generate a single front? In plant roots, each trichoblast cell polarizes to grow a single root hair (*Cole and Fowler, 2006*). Root hair outgrowth is regulated by the GTPase Rop2, a member of a plant-specific 'Rop' family closely related to Cdc42 and Rac GTPases (*Jones et al., 2002*). Strikingly, mutations in a plant GDI gene lead to the frequent production of multiple growing root hair sites in a single cell (*Carol et al., 2005*). We speculate that competition between polarity sites for Rop2 may ensure that only one root hair grows per cell. By analogy to our findings for yeast, slowing the exchange of Rop2 may impair competition in that system, allowing more than one site to initiate tip growth.

## Materials and methods

### Yeast strains

All yeast strains (listed in *Table 1*) are in the YEF473 background (*his3-Δ200 leu2-Δ1 lys2-801 trp1-Δ 63 ura3-52*) (*Bi and Pringle, 1996*) or BF264-15Du background (*ade1 his2 leu2-3,112 trp1-1 ura3Δns*) (*Richardson et al., 1989*). Deletion of *BNI1* was performed as described (*Chen et al., 2012*). The polarity markers Bem1-GFP (*Kozubowski et al., 2008*), Spa2-mCherry (made by the PCR-based C-terminal tagging method [*Longtine et al., 1998*]), Cdc3-mCherry (*Tong et al., 2007*) and Abp1-mCherry (*Howell et al., 2009*) replace endogenous genes and are functional. H2B-mCherry (a gift from Kerry Bloom) was amplified by PCR using genomic DNA as template and integrated at the endogenous locus. Whi5-tdTomato (a gift from Chao Tang) was integrated at the endogenous locus. The polarity markers GFP-Cdc42, Cdc42-mCherry[SW] and GFP-Sec4 (*Chen et al., 2012*) are integrated at the *URA3* locus. The GFP-Cdc42 marker contains a linker APPRRLVHP between the N-terminal GFP and Cdc42 to increase the functionality (*Kuo et al., 2014*). An integrating *URA3* plasmid containing Cdc42-mCherry[SW] was constructed following the methods from Sophie Martin's lab. First, a linker sequence (GGCTCTGGCAGATCTGCATGCTCTCTCGAGGCGGGCGGC) was introduced between leucine 134 and arginine 135 of *CDC42* on the plasmid. mCherry was then cloned into the *Bgl*II and *Xho*I sites on the linker sequence, leaving 5-amino acid linkers flanking mCherry. The resulting plasmid was targeted for integration at the *URA3* locus by cutting at the unique *Eco*RV site.

To generate Bem1-GFP-CAAX, a sequence (AAGAAAAGTAAGAAATGTGCCATCCTGTAA) encoding the polybasic-prenyl motif was introduced before the stop codon of GFP on an integrating *BEM1-GFP* plasmid. This plasmid was then targeted for integration at the *BEM1* locus by cutting at the unique *Pst*I site in *BEM1*. To generate Cdc24-GFP-CAAX and Cdc24-CAAX (as well as nonphosphorylatable derivatives (*Kuo et al., 2014*)), we constructed new vectors for PCR-based C-terminal tagging of genomic loci (*Longtine et al., 1998*): pFA6a-GFP(S65T)-CAAX and pFA6a-CAAX insert the same polybasic-prenyl motif.

All strains are in the YEF473 (*his3-△200 leu2-△1 lys2-801 trp1-△63 ura3-52*) or BF264-15Du (*ade1 his2 leu2-3,112 trp1-1 ura3Δns*) backgrounds.

### Live-cell microscopy

Cells were grown in synthetic medium (CSM) (MP Biomedicals, Santa Ana, CA) with 2% dextrose at 30°C. In order to image polarity establishment, we used a hydroxyurea arrest/release synchrony protocol that allows us to catch more cells at the time of polarization and also protects cells from phototoxic stress during imaging (*Howell et al., 2012*). Prior to imaging, cells were diluted to $5\times10^6$ cells/ml, arrested with 200 mM hydroxyurea (Sigma-Aldrich, St. Louis, MO) at 30°C for 3 hr, washed, released into fresh synthetic medium for 1 hr, harvested and mounted on a slab composed of

medium solidified with 2% agarose (Denville Scientific Inc., Holliston, MA). The slab was placed in a temperature-controlled chamber set to 30°C for imaging. Images were acquired with an Andor Revolution XD spinning disk confocal microscope (Olympus, Japan) with a Yokogawa CSU-X1 5000 r. p.m. disk unit, and a 100x/1.4 UPlanSApo oil-immersion objective controlled by MetaMorph software (Universal Imaging, Bedford Hills, NY). Images (stacks of 30 images taken at 0.24 µm z-steps or stacks to 15 images taken at 0.5 µm z-steps) were captured by an iXon3 897 EM-CCD camera with 1.2x auxiliary magnification (Andor Technology, UK). The laser power was used at 10% maximal output. An EM-Gain setting of 200 was used for the EM-CCD camera. Exposure to the 488 nm and 561 nm diode lasers was 200 ms.

To compare the whole cell intensities or peak intensities of polarized foci between strains, two strains were mixed in a 1:1 ratio and put on the same slab for imaging. Strain identity was distinguished using either a unique marker (e.g. Spc42-mCherry) or brief prestaining with fluorescent concanavalin A (Life Technologies, Carlsbad, CA) (*Lew and Reed, 1993*).

Scanning confocal images were acquired with a Zeiss 780 confocal microscope with an Argon/2 and 561nm diode laser, a 63x/1.4 Oil plan-Apochromat 44 07 62 WD 0.19 mm objective, and captured with a GaAsP high QE 32 channel spectral array detector using Zen 2010 software (Carl Zeiss, Germany). Representative cells were assembled for presentation using ImageJ (FIJI) and Illustrator (Adobe, San Jose, CA).

## Latrunculin A or rapamycin treatment

Cells were grown to mid-log phase in CSM + dextrose overnight at 24°C, mounted onto agarose slabs containing the same medium with 200 µM Latrunculin A (Life Technologies, Carlsbad, CA) or 50 µg/ml rapamycin or DMSO (control) and imaged.

## Fluorescence recovery after photobleaching

Exponentially proliferating cells were mounted on a 2% agarose slab and imaged on a DeltaVision Elite microscope (GE Healthcare Life Sciences, UK) with a 100x/1.40 oil UPLSAPO100XO objective, an InsightSSI[TM] Solid State Illumination source, and an outer temperature control chamber set to 30°C. Photobleaching of a polarized focus was performed using the Photokinetics function in the SoftWoRx 5.0 software (Applied Precision, Slovakia) with one iteration, 0.1 s bleaching at 10% power of a 488 laser. Three images were acquired before the bleaching event and the fluorescence recovery after photobleaching was monitored by 23 image acquisitions with adapted time intervals. Images were captured using an Evolve[TM] 512 back-thinned EM-CCD camera (Photometrics, Tucson, AZ) with an EM gain of 200. 2% transmission of the light source was used to illuminate cells. Exposure was 250 ms for Bem1-GFP, Bem1-GFP-CAAX and GFP-Cdc42 probes.

FRAP analyses were performed on unbudded cells with a strong polarized focus. The bleach zone encompassed a circular region around the polarized focus with ~1 µm diameter. Changes in fluorescence intensities in the bleach zone were measured by MetaMorph, and after background intensity subtraction the signal was normalized to the pre-bleaching value. Normalized data were not well fitted by a single exponential, presumably because recovery of bleached cytoplasm within the circular region occurred on a rapid timescale relative to recovery of the membrane signal. Thus, curves were fitted with a double exponential model in MATLAB (Mathworks, Natick, MA), and the recovery half-time was calculated using the slower exponential rate constant.

## Fluorescence loss in photobleaching (FLIP)

The microscopic settings for FLIP experiments were the same as for FRAP except that the bleaching event was performed with 200 ms laser duration and the exposure was 500 ms for Bem1-GFP and Bem1-GFP-CAAX probes. Cells were imaged once pre-bleach, followed by 35 iterations of bleaching and imaging events at approximately 0.5 s (Bem1-GFP) or 5 s (Bem1-GFP-CAAX) intervals. FLIP analyses were performed on unbudded cells with a strong polarized focus. The bleach zone encompassed a circular region with ~1 µm diameter in the cytoplasm away from the focus. Fluorescence intensities were measured by MetaMorph. In addition to measuring the intensity at the polarity focus, fluorescence intensity in a neighboring cell was measured to correct for indirect bleaching. Changes in fluorescence intensities were calculated by $(\text{Intensity}_{polarity\_focus} - \text{Intensity}_{background})/$

(Intensity$_{neighbor}$− Intensity$_{background}$) and plotted against time. For the Bem1-GFP-TM probe, which does not polarize, fluorescence loss was measured at a patch on the plasma membrane.

## Deconvolution, image analysis, and quantification

For timelapse series, images were deconvolved using Huygens Essential software (Scientific Volume Imaging, Netherlands). The classic maximum-likelihood estimation and predicted point spread function method with signal-to-noise ratio 3 was used with a constant background across all images from the same channel on the same day. The output format was 16-bit, unscaled images to enable comparison of pixel values.

To detect polarity foci in different focal planes, maximum intensity projections were constructed and scored visually for the presence of more than one focus. The coexistence time is the interval between the first frame in which more than one spot was detected and the frame when only one spot was detected.

To quantify probe intensities in two-color movies, we developed a MATLAB-based Graphical User Interface (GUI) named *Vicinity*. The GUI displays time-lapsed imaging records of summed-projection z-stacks (as TIFF stacks) from two fluorescence channels side by side (*Figure 12*), identifies and tracks polarity spots in one of the channels (Bem1), and measures intensity levels of both markers in the vicinity of these spots. The vicinity of a polarity spot is defined as a circular region centered on the spot centroid. The radius of the circle is specified by the user. Image processing by *Vicinity* consists of the following steps (*Figure 12*):

1. Two threshold values are specified interactively with a slider: the lower value is used to separate cells from the background and the higher value is used to define polarity spots within the cells.
2. The radius of the circular regions and the 'filter size' (in square pixels) need to be specified. The filter size defines the minimum spot size to be considered. Specifying a non-zero filter size allows the exclusion of small random spots that appear due to noise.
3. Vicinity then detects and tracks all spots satisfying the above user-specified criteria. Our automatic tracking algorithm is based on finding the nearest spot at time $t+1$ within the region of a user-specified radius ('target size') around the centroid of each spot at time $t$. If a spot temporarily disappears (blinks) due to intensity fluctuations, *Vicinity* can keep tracking the spot if the 'remove blinking' option is on.
4. The user can choose any of the tracks for quantitation. The mean (or max) intensity values of all non-background pixels in the vicinity of the tracked spots are displayed as a function of time for both channels side by side. The measurements for multiple spots can be added to the query and saved as a data file in text format for further statistical analysis.

A threshold was set that would only select the polarized Bem1 focus. The centroid of the Bem1 focus was then used to define a circular region covering the polarity site. The mean pixel intensities within the circular region for both green and red channels were calculated and the corresponding background intensities (determined from one time frame before the polarization signal was detected) were subtracted. Changes in intensity were reported as percent of maximum (sum of all polarized foci) within the period of interest for that cell.

Quantification of cortical to cytoplasmic fluorescence for Bem1-GFP and Bem1-GFP-CAAX probes was performed as described previously (*Kuo et al., 2014*).

To compare the whole cell fluorescence intensities or peak intensities of polarized foci between two strains in a mixed-cell experiment, the raw images were denoised with the Hybrid 3D Median Filter plugin in ImageJ (http://rsb.info.nih.gov/ij/plugins/hybrid3dmedian.html) and quantified using Volocity (PerkinElmer, Waltham, MA).

To measure Bem1-GFP intensity at the patch relative to whole cell fluorescence before bud emergence, a threshold was set to determine both total patch and cell fluorescence for each cell and quantified using Volocity.

To quantify whole-cell fluorescence, a constant threshold was set across all the stage positions on the same slab that selected the entire cell. The mean pixel intensity of each cell was normalized to the average of the control strain. Quantification of peak intensities was similar except that the threshold was set to select only the polarized foci and the peak value within the polarization period was picked out for normalization. Images were processed for presentation using MetaMorph and ImageJ.

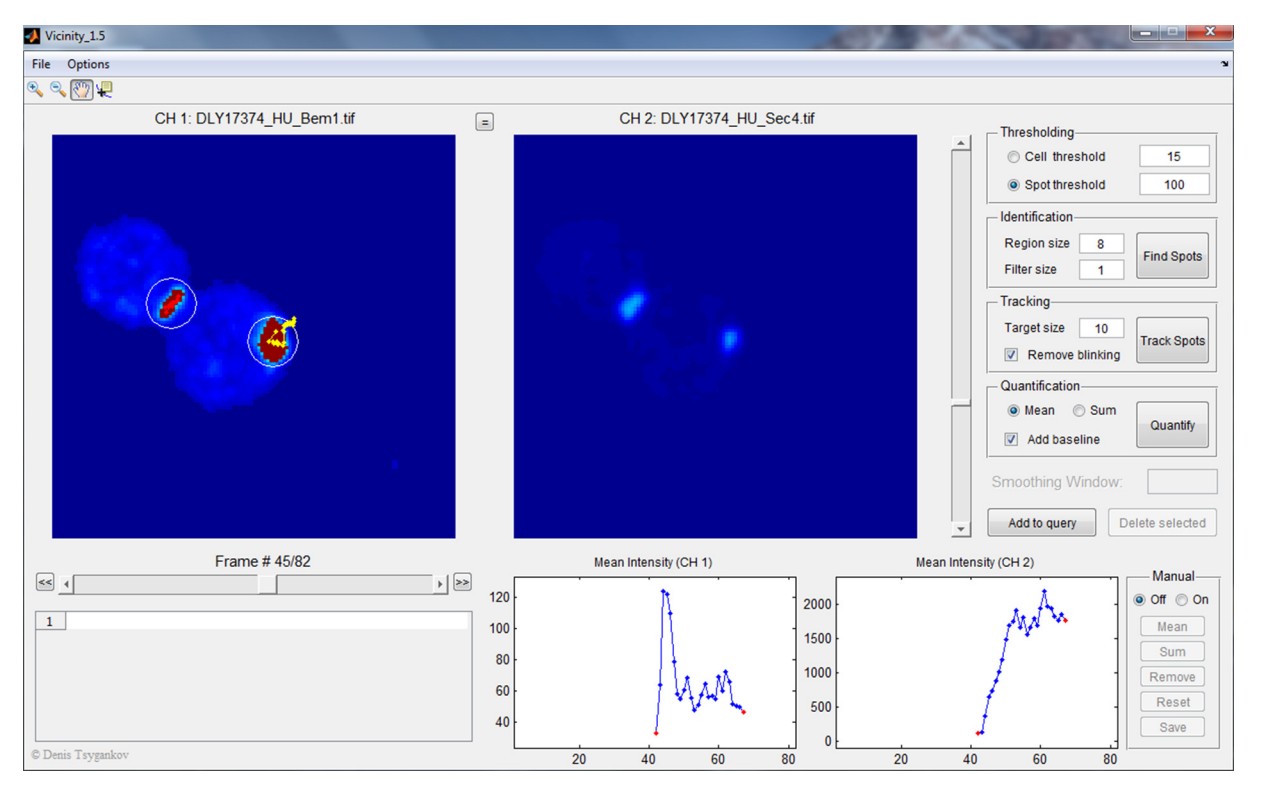

**Figure 12.** Screenshot illustrating *Vicinity* GUI operation. The upper left of this interface shows the sum projection z-stacks from two fluorescence channels (Bem1-tdTomato and GFP-Sec4 in this case) side by side. The upper right side is the control panel where the threshold for selecting cells and polarity spots, radius of circular regions, filter size, and target size are set. Users can choose to quantify either mean or sum intensity of the pixels in the circular regions. In this example, both the polarity spot and the neck signal were marked in circular regions because their intensities were above the spot threshold, but only the polarity spot was selected for quantification (the track was highlighted in yellow). The intensity changes in the selected region over time (in both channels) are reported in the lower right side of the interface.

## Immunoblots

$10^7$ cells were collected for each sample and total protein was extracted by TCA precipitation as described (*Keaton et al., 2008*). Electrophoresis and Western blotting were performed as described (*Bose et al., 2001*). Monoclonal mouse anti-Cdc42 antibodies (*Wu and Brennwald, 2010*) were used at 1:500 dilution. Monoclonal mouse anti-GFP antibody (Roche Applied Science, Germany) was used at a 1:1000 dilution. Polyclonal rabbit anti-Cdc11 antibody (Santa Cruz Biotechnology, Dallas, TX) was used at a 1:5000 dilution. Fluorophore-conjugated secondary antibodies against mouse (IRDye 800CW goat anti-mouse IgG, LI-COR Biosciences, Lincoln, NE) or rabbit (Alexa Fluor 680 goat anti-rabbit IgG, Invitrogen, Carlsbad, CA) antibodies were used at 1:5000 dilutions. Blots were visualized and quantified using the ODYSSEY imaging system (LI-COR Biosciences).

## Computational methods: analysis of competition

Analysis of competition was performed using a model adapted from (*Goryachev and Pokhilko, 2008*), diagrammed in *Figure 11A*. Membrane-localized Cdc42 exchanges between GDP-bound and GTP-bound states. GDP/GTP exchange is catalyzed by the GEF, in complex with Bem1. This complex exchanges between membrane and cytoplasm, and can bind reversibly to GTP-Cdc42. Two other Cdc42 regulators are represented implicitly by first-order reactions: GAPs promote GTP hydrolysis by Cdc42 (rate constant k2b), and the GDI reversibly exchanges GDP-Cdc42 between membrane and cytoplasm (rate constants k5a and k5b). Positive feedback occurs because binding of the Bem1 complex to GTP-Cdc42 increases local GEF activity in regions with higher GTP-Cdc42,

generating more local GTP-Cdc42, which can in turn recruit more Bem1 complex. The equations are deterministic with the exception of the Bem1 complex, which is subject to Gaussian white noise $\xi(t,x)$ with the strength s = 0.0001, as follows:

$$\frac{\partial Cdc42T}{\partial t} = (k_{2a}BemGEF_m + k_3 BemGEF42) \cdot Cdc42D_m - k_{2b}Cdc42T$$
$$- (k_{4a}BemGEF_m + k_7 BemGEF_c) \cdot Cdc42T + k_{4b}BemGEF42 + D_m\Delta Cdc42T$$

$$\frac{\partial BemGEF42}{\partial t} = k_{2b}Cdc42T - (k_{2a}BemGEF_m + k_3 BemGEF42) \cdot Cdc42D_m$$
$$- k_{5b}Cdc42D_m + k_{5a}Cdc42D_c + D_m\Delta Cdc42D_m$$

$$\frac{\partial Cdc42D_m}{\partial t} = (k_{4a}BemGEF_m + k_7 BemGEF_C) \cdot Cdc42T - k_{4b}BemGEF42 - D_m\Delta BemGEF42$$

$$\frac{\partial BemGEF_m}{\partial t} = k_{1a}BemGEF_c + k_{1b}BemGEF_m + k_{4b}BemGEF42$$
$$- k_{4a}BemGEF_m \cdot Cdc42T - \sqrt{s}\xi(t,x) + D_m\Delta BemGEF_m$$

$$\frac{\partial Cdc42Dc}{\partial t} = \frac{\eta}{A}\int (k_{5b}Cdc42D_m - k_{5a}Cdc42D_c)dA$$
$$\frac{\partial BemGEF_c}{\partial t} = \frac{\eta}{A}\int \left(k_{1b}BemGEF_m - k_{1a}BemGEF_c - k_7 BemGEF_c \cdot Cdc42T + \sqrt{s}\xi(t,x)\right)dA$$

The equations were discretized and solved on a square uniform grid with periodic boundary conditions, generating a torus. All membrane species have the same diffusion coefficient. The cytoplasm is assumed to be well mixed, approximating fast cytoplasmic diffusion. Parameter values are listed in *Table 2*. These have evolved since the original model (*Goryachev and Pokhilko, 2008*) for a variety of reasons including new biochemical measurements (*Howell et al., 2009*), adjustments to fit in vivo data (*Savage et al., 2012*), and recognition of negative feedback (*Howell et al., 2012*; *Kuo et al., 2014*). To keep the model tractably simple, we did not consider negative feedback in our analysis. Instead, we raised the GAP activity to keep the peak size realistic even without negative feedback.

To simulate competition, we began with the homogeneous steady state and provided two identical perturbations at diametrically opposite locations, leading to the growth of two identical peaks and concurrent partial depletion of Cdc42 and Bem1 complexes from the cytoplasm (*Video 9*). At this unstable steady state, each peak maintains a dynamic balance of recruitment and loss of Cdc42 and Bem1 complexes. Continued simulation with noise yielded a minuscule difference between peaks, initiating the growth of one peak at the expense of the other (*Video 9*) (*Figure 9C*). During most of this 'competition' phase, cytoplasmic levels of Cdc42 and Bem1 complexes remained stable (*Figure 9D*). During competition, we tracked the net rates of recruitment of Cdc42 and Bem1 complexes from the cytoplasm, and the Cdc42 fluxes are plotted as a function of total Cdc42 amount within the peak in *Figure 9J*. Note that net fluxes from the cytoplasm can be positive even for losing peaks: the losing peak nevertheless shrinks because these fluxes are no longer sufficient to combat loss via diffusion. Towards the end of the competition, the winning peak grew further and cytoplasmic concentrations decreased (*Figure 9D*), leading to a reduced net flux from the cytoplasm to the peak (*Figure 9J*).

The recruitment rate of Bem1-GEF complexes from the cytoplasm to the center of a polarity peak by active Cdc42 (Cdc42T) is given by $k_7 \cdot BemGEF_c \cdot Cdc42T$. Therefore, for a fixed amount of cytoplasmic Bem1-GEF complex the recruitment rate grows linearly with active Cdc42 (*Figure 9E*).

To determine the rate at which Bem1-GEF complexes are lost from the center of a polarity peak to the cytoplasm, we simulated the rate equations based on the reactions shown in *Figure 9F* (cartoon inset) with an initial Bem1-GEF concentration of 70 $\mu$M and GTP-Cdc42 levels ranging from 0 to 450 $\mu$M. The half time of Bem1-GEF was extracted from the simulations (*Figure 9G*).

**Table 2.** Parameters of the model.

| Description | Parameters | Value | Units | Reference |
|---|---|---|---|---|
| BemGEF$_c$ -> BemGEF$_m$ | $k_{1a}$ | 10 | s$^{-1}$ | Kuo et al., 2014 |
| BemGEF$_m$ -> BemGEF$_c$ | $k_{1b}$ | 10 | s$^{-1}$ | Kuo et al., 2014 |
| BemGEF$_m$ -> BemGEF$_c$ (Gaussian Noise) | s | 0.0001 | s$^{-1}$ | Kuo et al., 2014 |
| Cdc42D$_m$ + BemGEF -> Cdc42T | $k_{2a}$ | 0.16 | μM$^{-1}$ s$^{-1}$ | Kuo et al., 2014 |
| Cdc42T -> Cdc42D$_m$ | $k_{2b}$ | 1.75 | s$^{-1}$ | This study |
| Cdc42D$_m$ + BemGEF42 -> Cdc42T | $k_3$ | 0.35 | μM$^{-1}$ s$^{-1}$ | Kuo et al., 2014 |
| BemGEF + Cdc42T -> BemGEF42 | $k_{4a}$ | 10 | μM$^{-1}$ s$^{-1}$ | Kuo et al., 2014 |
| BemGEF42 -> BemGEF + Cdc42T | $k_{4b}$ | 10 | s$^{-1}$ | Kuo et al., 2014 |
| Cdc42D$_c$ -> Cdc42D$_m$ | $k_{5a}$ | 36 | s$^{-1}$ | Kuo et al., 2014 |
| Cdc42D$_m$ -> Cdc42D$_c$ | $k_{5b}$ | 0.65 | s$^{-1}$ | Kuo et al., 2014 |
| BemGEFc + Cdc42T -> BemGEF42 | $k_7$ | 10 | μM$^{-1}$ s$^{-1}$ | Kuo et al., 2014 |
| Diffusion coefficient on the membrane | $D_m$ | 0.0025 | μm$^2$ s$^{-1}$ | Kuo et al., 2014 |
| Membrane to cytoplasm volume ratio | $\eta$ | 0.01 | | Kuo et al., 2014 |
| Surface area of the membrane | A | 25π | μm$^2$ | Kuo et al., 2014 |
| Total [Cdc42] | | 1 | μM | Kuo et al., 2014 |
| Total [BemGEF] | | 0.017 | μM | Goryachev, 2008 |

If we apply a quasi-steady-state approximation to the fast reactions governing the binding and release of GTP-Cdc42 from the Bem1-GEF complex, we have:

$$k_{4a} BemGEF_m \cdot Cdc42T \approx k_{4b} BemGEF42$$

Thus, for a given Cdc42T, the concentration of Bem1-GEF in the center of the peak is:

$$BemGEF = BemGEF_m + BemGEF42 = BemGEF_m \cdot \left( 1 + \frac{k_{4a}}{k_{4b}} Cdc42T \right)$$

And the Bem1-GEF concentration changes according to:

$$\frac{dBemGEF}{dt} = -k_{1b} BemGEF_m = -\frac{k_{1b} k_{4b}}{k_{4b} + k_{4a} Cdc42T} BemGEF$$

The above equation is a first order reaction with an effective rate constant dependent on the active Cdc42 amount. Therefore, curves showing the time-dependent loss of Bem1-GEF (*Figure 9F*) can be fitted by exponential decay curves, the half time of which increases linearly with GTP-Cdc42 (*Figure 9G*):

$$T_{\frac{1}{2}} = \frac{ln2}{k_{effective}} = ln2 \cdot \left( \frac{k_{4a}}{k_{1b} k_{4b}} Cdc42T + \frac{1}{k_{1b}} \right)$$

To determine the dwell time for Cdc42, we considered only GAP-mediated GTP hydrolysis and the competing GEF and GDI reactions (*Figure 9H*, cartoon inset). We calculated the loss of Cdc42 (initial concentration 300 μM) with time for different Bem1-GEF-Cdc42 concentrations exactly as we did for Bem1-GEF, and plotted the resulting dwell times for varying GEF concentration (*Figure 9H*).

If we apply a quasi-steady-state approximation to the exchange between GDP-Cdc42 and GTP-Cdc42, we have:

$$k_3 BemGEF42 \cdot Cdc42D_m \approx k_{2b} Cdc42T$$

And the Cdc42 concentration changes according to:

$$\frac{dCdc42}{dt} = -k_{5b} Cdc42D_m = -\frac{k_{2b} k_{5b}}{k_{2b} + k_3 BemGEF42} Cdc42$$

Thus, the half time increases linearly with GEF:

$$T_{\frac{1}{2}} = ln2 \cdot \left( \frac{k_3}{k_{2b}k_{5b}} BemGEF42 + \frac{1}{k_{5b}} \right)$$

To estimate the loss of Cdc42 from a polarity peak by lateral diffusion (*Figure 9I*), we began with the concentration profiles of the winning and losing peaks from the full simulation (*Video 9*). The total Cdc42 content within the waistline was normalized to the content in the final winning peak (X axis). The rate of loss of Cdc42 by diffusion across the waistline was divided by the Cdc42 content within the waistline for each peak to derive a% loss/s measure (Y axis).

## Computational methods: Linear stability analysis

Linear stability analysis (LSA) was performed following the method of (*Klunder et al., 2013*). Here we provide a brief summary of the procedure. A full description of the model and details of the method appear in the Supplemental Information of the original paper. A diagram of the model is presented in *Figure 11B*. The model consists of 4 membrane bound species: GTP-Cdc42, GDP-Cdc42, Bem1, and Bem1-Cdc24 complex; and 3 cytosolic species: Cdc42-GDP, Bem1, and Cdc24.

LSA is used to determine when the spatially homogenous solution to the model equations becomes unstable to infinitesimally small perturbations. The first step in the process is to linearize the model equations around the homogenous solution. The linear equations govern the system's response to small perturbations and can be used to determine which spatial modes become unstable as a model parameter is varied. Because the computational domain is a sphere, solutions to the linearized equations can be represented as a series solution in terms of spherical harmonics and a modified Bessel function of the first kind. The eigenvalues associate with the modes (l,m) of the spherical harmonic expansion satisfy characteristic equations determined by the model equations and boundary conditions. We numerically find the roots of the characteristic equations and look for eigenvalues that have positive real parts. Eigenvalues with positive real parts indicate exponential growth of that mode and are a sufficient condition for demonstrating the homogenous solution is unstable. A necessary condition for competition between peaks is the existence of more than one eigenvalue with positive real part.

We first reproduced the published results (*Klunder et al., 2013*) to verify our numerical methods (*Figure 11C–E*). We then repeated the analysis for cases in which: 1) the molecular abundance of all components was increased 5.86-fold (258/44) to account for the increased volume of the model sphere (258 fL) compared to the average haploid cell (44 fL) (*Figure 11C*), 2) the Cdc42 diffusion coefficient was varied between 0.03 µm²/s and 0.0025 µm²/s (*Figure 11D*) and 3) both the Cdc42 abundance and diffusion coefficient were varied (*Figure 11E*). In each case, our analysis revealed multiple eigenvalues with positive real parts suggesting the existence of competition between polarity factors.

## Acknowledgements

We thank Felipe Bendezu and Sophie Martin (UNIL, Switzerland) for sharing information on *S. pombe* Cdc42-mCherry[sw] prior to publication. We thank Steve Haase, Nick Buchler, Stephano Di Talia, and members of the Lew lab for comments on the manuscript and many stimulating discussions. NSS was supported by a Wellcome Trust ISSF Non-Clinical Fellowship. This work was funded by NIH/NIGMS grants GM62300 to DJL, and GM103870 and GM079271 to DJL and TCE.

## Additional information

### Funding

| Funder | Grant reference number | Author |
| --- | --- | --- |
| National Institute of General Medical Sciences | R01 GM62300 | Daniel J Lew |
| National Institute of General Medical Sciences | R01 GM103870 | Timothy C Elston Daniel J Lew |

| National Institute of General Medical Sciences | R01 GM079271 | Timothy C Elston Daniel J Lew |
| Wellcome Trust | ISSF Non-clinical Fellowship | Natasha S Savage |

The funders had no role in study design, data collection and interpretation, or the decision to submit the work for publication.

## Author contributions

CFW, JGC, BW, Conception and design, Acquisition of data, Analysis and interpretation of data, Drafting or revising the article; MM, NSS, TCE, DJL, Conception and design, Analysis and interpretation of data, Drafting or revising the article; DT, Analysis and interpretation of data, Drafting or revising the article, Contributed unpublished essential data or reagents; TRZ, Generated yeast strains, Conception and design, Contributed unpublished essential data or reagents

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
