## [Decision Letter]

[Editors’ note: a previous version of this study was rejected after peer review, but the authors submitted for reconsideration. The previous decision letter after peer review is shown below.]

Thank you for choosing to send your work entitled "Role of competition between polarity sites in establishing a unique front" for consideration at *eLife*. Your full submission has been evaluated by Vivek Malhotra (Senior editor), a member of the Board of the Reviewing Editors and two peer reviewers with expertise in modelling as well as cell polarity, and the decision was reached after discussions between the reviewers and the editors. Based on our discussions and the individual reviews below, we regret to inform you that your work will not be considered further for publication in *eLife*.

While the reviewers and editors acknowledge the importance of the topic they remain concerned about the design of the study itself (articulated by Reviewer 1) and the extent of advance from the computational analyses (as discussed by Reviewer 2).

*Reviewer #1:*This manuscript deals with the mechanism that ensures formation of a single polarized cell front. Positive feedback mechanisms are known to promote spontaneous polarization by enhancing stochastic fluctuations of Cdc42 at the cell cortex. Previous observations from the same group showed that, in budding yeast, the emergence of polarized sites can occur at multiple sites simultaneously, but then evolves into a single site. These observations suggest that polarity sites are in competition with each other. However, the underlying mechanism of this competition for singularity is not well understood. Here, the authors propose that competition is governed by the dynamic turnover of polarity proteins at the polarized site.

While the question is very interesting, the manuscript falls short of demonstrating the mechanism behind polarity sites competition. I believe the main issue is a conceptual flaw in the design of the study, described below.

To slow down Bem1 and Cdc24 (a scaffold and GEF part of a positive feedback mechanism on Cdc42, respectively) exchange between membrane and cytosol, the authors choose to add a prenylation CAAX box (from Cdc42). Can prenylated proteins readily detach and re-attach at the membrane? For Cdc42, extraction from the membrane requires a dedicated GDI (and indeed their own model does not hypothesize any attachment/detachment of Cdc42-GTP, which is not a substrate (or a poor substrate) for the GDI). The authors state that "Bem1-CAAX exchange between membrane and cytosol was slower than that of Bem1". However, I do not see any evidence that Bem1-CAAX is cytosolic at all. The internal signal shown in Figure 3 may also be due to endo-membrane association. In addition, as prenylation occurs post-translationally at the ER, prenylated proteins are trafficked through the secretory system. Thus, Bem1-CAAX may well be delivered and retrieved from the plasma membrane via exo- and endocytosis. This allele may thus not be too dissimilar from the Bem1-Snc2 fusion previously published by this group, which also led to multi-bud formation, but was interpreted to do so because it used the slower actin-based feedback system (Howell, Cell 2009). Because of its enhanced membrane binding, the Bem1-CAAX allele certainly also has effects on the levels of available Bem1 (even if global levels are unchanged) and on Bem1 diffusion rates at the membrane. I thus do not think that this approach allows them to specifically slow down membrane-cytosol exchange, nor make any strong conclusion about it.

Reviewer #1 (Minor Comments):

Most or all of the study appears to be conducted in an *rsr1∆* background, though this is not stated. This should be made explicit.

The first part of the manuscript investigates candidate proteins that may be involved in stabilizing the polarity patch, and concludes that neither actin nor septins appear to have this function. Though I would agree with this conclusion, the description of the septin data seems inaccurate. In Video 4, there is a weak Cdc3 signal on the disassembling left patch at timepoint 8:15, and one before the transient disassembly of the right patch at 9:45 to 11:15, in agreement with recent data that septins contribute in negative feedback for Cdc42 patch disassembly (see for instance Okada et al. Dev Cell 2013). Thus, septins may well contribute to patch destabilization in addition to any competitive mechanism.

The data on *rdi1∆* slowing down Cdc42 exchange and producing multi-buds reproduces previously published data (see Slaughter, Dev Cell 2009; Freisinger, NatComm 2013). This data could be provided as supplement.

I am bothered by the fact that the authors explain at the beginning of their manuscript that GFP-Cdc42 is poorly functional (and thus justify use Bem1-GFP instead), but use this allele to quantify Cdc42 turnover without any disclaimer. In Figure 1, they use a Cdc42-mCherry^SW^ allele modeled on a recently published functional allele in *S. pombe* (Bendezu et al., PLoS Biol 2015; which should be cited), which they show in Figure 9 is not fully functional in *S. cerevisiae*. This information should be better integrated in the text. As it stands, Figure 9 is only referenced in the Materials and methods section (and this also applies to Figure 10). Also, the Cdc42-mCherry^SW^ strain construction is not entirely clear. This construct is said in the Materials and methods to be integrated at the URA3 locus (under which promoter is not stated), but is said in the legend to Figure 9 to replace endogenous cdc42. This should be clarified.

The reasons for the poor viability of the Cdc24-CaaX allele are unclear. Previous work showed that a myristoylated allele of Cdc24, also targeted to the membrane, grows well even in absence of *rsr1* (Shimada et al., EMBO 2004).

The Bem1-Caax localization in Figure 4 looks very different from Figure 3. Why?

I am confused by the limiting or non-limiting nature of polarity factors and the statements made around this by the authors. They first state that "the ability of cells to polarize simultaneously in up to four directions without any overexpression indicates that polarity factors and other components required for budding are present at higher levels than needed to make one bud". I am not sure I understand how competition works if the substrate(s) for competition is not limiting. In the multi-site, multi-budded cells, is the local abundance of the polarity factors reduced compared to the single-site, single-budded situation?

I am further confused by how the statement "because of positive feedback, stochastic activation of a small amount of Cdc42 somewhere on the membrane leads to further accumulation of active Cdc42 until depletion of the cytoplasmic pools of polarity proteins stops the process" can be reconciled with the statement in the next paragraph that "the uniqueness of the Cdc42 peak is not due to insufficient Cdc42 or Bem1-Cdc24". Does the drop in cytosolic Bem1 after competition resolution predicted in Figure 7 really happen? This should correlate with a local increase of Bem1 at the membrane. But Figure 3 showed an overshoot of Bem1 during competition and lower level after.

One conclusion from the data is that the reduced competition observed in Bem1-CaaX and *rdi1∆* conditions does not qualitatively alter the model, but quantitatively changes parameters and thus the timescale of competition. This would suggest that the multi-budding phenotype could be rescued by delaying G1-S transition until resolution of the polarity patches to a single one. Is that the case?

In the final part of the manuscript, the authors aim to resolve a difference between the output of their model, which yields multiple initial clusters resolving into one, and the model from Klunder et al., 2013, which showed a single initial broad peak sharpening over time, and conclude that the differences are due to parameter value choices. This is an interesting reconciling finding, but their choice of parameters for Cdc42 lateral diffusion is low. The value chosen, 0.0025µm2/s was measured for transmembrane proteins. More recent values for Cdc42 were obtained, ranging from 0.013 µm2/s to 0.0061 µm2/s in *S. cerevisiae* (Marco et al., Cell 2007 and Freisinger et al., NatComm 2013) and up to 10-fold higher using functionally tagged Cdc42 in *S. pombe* (Bendezu et al., PloS Biol 2015).

Finally, given most the experiments are done modifying Bem1 properties, it would be necessary to test the effect of limiting Bem1 levels and thus constructing Bem1 hemizygote diploids.

*Reviewer #2:* Wu et al. test two alternative hypotheses on the regulation of the establishment of a unique polarity site in budding yeast cells. They conclude that a competition between different sites rather than a stabilizing factor is responsible for the collapse of all but one polarity zone. They test the corresponding mathematical models of the two hypotheses and find that their earlier published model explains the underlying mechanism. Results of Figure 5 are convincing that a growing bud can be outcompeted and turned off, disproving the stabilizer hypothesis. The comparison of the computational models supports this conclusion, but does not go beyond earlier explanations of the system.

The conclusion that the competition mechanism works through positive feedback loops without an inhibitor tells me that it works with substrate-depletion. The sentence in the second paragraph of the subsection “Mechanism of competition in a computational model” does not support this idea, but as Bem1 overexpression can also lead to multi-budded cells and its level goes down on Figure 7 assume Bem1 is (at least part of) the limiting substrate in this Turing-like system. If we follow the theory on such systems (Segel, L. A., and Jackson, J. L. (1972) Journal of Theoretical Biology, 37(3), 545-559), then not only changes in diffusion rate, but also changes in the size of cells could have similar effects as the critical wavelength of the underlying Turing mechanism depends on both. Thus it would support the conclusions if the authors could show that cells with larger size (e.g. HU block and release) can form multiple peaks, especially if combined with the slow diffusing membrane bound proteins. From their current data they might be already able to analyze if there is a correlation between cell size and number of appearing buds in the strains investigated in Figure 6.

If it is a competition mechanism with positive feedbacks in a Turing-like system then a gradient of the limiting compound could be shown. Probably by expressing in low level, an extra copy of Bem1-GFP could show an actual gradient around polarity zones and the change in size of these zones during competition.

The authors chose to drop the negative feedback loop from their last model (Kuo et al. 2014) and replace with a constant high GAP activity. This indeed allowed them to explain the issue with the uniqueness of the final peak, but does not explain how the single peak disappears sometimes. Furthermore, the choice of dropping a potential inhibitor restricted the Turing-type of explanation to the substrate-depletion model. Probably both work better than the stabilizer model, but I feel this as a step back from their earlier results. Indeed it is not clear what new we learned from the model compared to earlier analysis of various versions of it.

[Editors’ note: what now follows is the decision letter after the authors submitted for further consideration.]

Thank you for resubmitting your work entitled "Role of competition between polarity sites in establishing a unique front" for further consideration at *eLife*. Your revised article has been favorably evaluated by Vivek Malhotra (Senior editor), Mohan Balasubramanian (Reviewing editor), and two reviewers. The manuscript has been improved but there are some remaining issues that need to be addressed before acceptance, as outlined below:

The reviewers appreciated the careful and considered response to the comments, including through the addition of a number of new experiments.

The reviewers have raised a few points, most of which can be addressed through rewriting, but if experiments are handy, I would like you to address the last major point raised by Reviewer 2 of targeting Cdc24 to the membrane through a means different from CAAX to peripherally anchor Cdc24 to the PM through new experiments or discuss this point clearly through rewriting in the revision.

With these revisions, your paper will make a really nice contribution to the field of cell polarity.

Please see the reviewers’ comments verbatim below and I will appreciate your response to these.

*Reviewer #1:*I was happy to see that the authors agreed that the proposed competition results from a substrate-depletion mechanism and I can see that their analysis identified the molecular mechanism that is standing behind the uniqueness of active polarity site selection in budding yeast.

Several earlier modelling papers investigated such systems (some summarized in Jilkine and Edelstein-Keshet 2011 PLoS Comput Biol), thus the computational finding is not really surprising from the theoretical point of view, but together with the experimental validation the manuscript has merit. I have only minor suggestions for possible further tests of the limiting factor hypothesis.

*Reviewer #1 (Minor Comments):*The authors cite the paper by Otsuji et al. claiming that any initial pattern could end up with a single peak in a substrate-depletion reaction system, but as they correctly also add, this was shown only for mass-conserved Turing systems. They claim that their system is mass-conserved, which might be true for the time window they investigate the cells (< 15 mins). The simulations were initiated with identical perturbations on two sites and followed what steady state is reached. To further validate the limiting factor idea it would be interesting to test what happens as the cells grow. There are several papers (by Crampin and Maini) on the behaviour of the Turing pattern in growing domains. I was wondering how the proposed model would behave if growth is considered.

A related question could be if and how the model could explain the periodic budding of clb1-6del cells. Could the increase in the concentration of the limiting factor induce a second tip to assemble?

In general the point I would like to see (at least by the model) is that the second tip can be formed not only for slowed down competition, but also if the second zone is above a given distance from the first or the limiting factor is not limiting anymore. These could further prove the limiting factor hypothesis.

*Reviewer #2:*My main concern, which was that addition of a prenyl group would lock the proteins on the membrane, has been largely addressed, but I would encourage the authors to slightly extend their work on Cdc24, as suggested below.

For Bem1-CAAX, the anchor-away experiment is convincing and indeed shows that Bem1-CAAX is mobile on and off the membrane. I would suggest that, in the flow of the paper, it would be better to describe this at the top of the Bem1-CAAX allele description as a way of introducing it as a tool to reduce exchange dynamics, rather than at the end. The conclusions made in the fourth paragraph of the subsection “Testing the competition model: reducing polarity protein” would then be more straightforward. In these lines, I would also suggest rephrasing the meaning of FLIP and FRAP quantification, as these do not directly show membrane-cytosol exchange dynamics, because lateral diffusion will also be a contributing factor. The proposed re-organization would also help in understanding the next paragraph, in which the logical link between the slowed membrane-cytosol exchange of prenylated protein and the previous analysis of Cdc42 dynamics is not entirely clear.

By contrast, the localization of Cdc24-CAAX to the nuclear membrane in haploid cells does not imply a detachment from the membrane. The nuclear membrane is connected to the ER, and it is possible that the relocation occurs as Cdc24-CAAX remains bound to the membrane rather than transit through the cytosol. Thus, the manuscript has been significantly improved and now convincingly shows that Bem1-CAAX shuttles between membrane and cytosol, but the same level of confidence does not apply to Cdc24-CAAX, as it would be predicted to be nuclear, rather than at the nuclear periphery in haploids if that were the case. I would feel much more comfortable if the data was confirmed using an alternative mode of binding these proteins (or at least Cdc24) peripherally to the plasma membrane, for instance through an amphipathic helix, a C2 or a PH domain.

---

## [Author Response]

[Editors’ note: the author responses to the first round of peer review follow.]

Reviewer #1: *While the question is very interesting, the manuscript falls short of demonstrating the mechanism behind polarity sites competition. I believe the main issue is a conceptual flaw in the design of the study, described below. To slow down Bem1 and Cdc24 (a scaffold and GEF part of a positive feedback mechanism on Cdc42, respectively) exchange between membrane and cytosol, the authors choose to add a prenylation CAAX box (from Cdc42). Can prenylated proteins readily detach and re-attach at the membrane? For Cdc42, extraction from the membrane requires a dedicated GDI (and indeed their own model does not hypothesize any attachment/detachment of Cdc42-GTP, which is not a substrate (or a poor substrate) for the GDI). The authors state that "Bem1-CAAX exchange between membrane and cytosol was slower than that of Bem1". However, I do not see any evidence that Bem1-CAAX is cytosolic at all. The internal signal shown in Figure 3 may also be due to endo-membrane association. In addition, as prenylation occurs post-translationally at the ER, prenylated proteins are trafficked through the secretory system. Thus, Bem1-CAAX may well be delivered and retrieved from the plasma membrane via exo- and endocytosis. This allele may thus not be too dissimilar from the Bem1-Snc2 fusion previously published by this group, which also led to multi-bud formation, but was interpreted to do so because it used the slower actin-based feedback system (Howell, Cell 2009).*

We fully agree that it is critical to show that Bem1−CAAX is not locked onto the membrane like Bem1−Snc2; if it were, then we would just have recapitulated the rewiring results of Howell 2009. We apologize for not making a convincing case for that, and thank the reviewer for catching the flaw. We have revised the paper significantly, and now show new data to bolster our case, as follows:

For Bem1−Snc2, polarization requires active endocytosis mediated by a Snc2 endocytosis motif (Howell et al. Cell2009) and actin−directed secretion on cables nucleated by the formin Bni1 (Chen et al. Mol. Biol. Cell2012). In contrast, Bem1−CAAX has no added endocytosis motif, and we now show that it does not require Bni1 (new Figure 5). Moreover, we show that Bem1−CAAX polarizes in the presence of Latrunculin A, implying that neither actin cable−directed secretion nor actin patch−dependent endocytosis is required (new Figure 5). Finally, using a rapamycin−induced dimerization “anchor away” strategy we show that Bem1−CAAX can become sequestered away from the polarity site through binding to ribosomes (new Figure 5), implying that it exchanges between membrane and cytoplasm.

We have not repeated these experiments for Cdc24−CAAX, but we believe that it behaves similarly. This is supported by Cdc24−CAAX localization in haploids. Unlike diploids, haploids express Far1, a protein that is important for mating and sequesters a pool of Cdc24 in the nucleus. Similarly, haploid but not diploid Cdc24−CAAX cells displayed Cdc24−CAAX enriched in the nuclear periphery (new Figure 5).

Neither Bem1−Snc2 nor Cdc24−Snc2 displayed this nuclear signal. We conclude that Bem1−CAAX and Cdc24−CAAX do exchange (albeit slowly) between the membrane and the cytoplasm, and do not polarize by the different vesicular mechanism employed by Bem1−Snc2.

We note that considerable published data from other labs supports the idea that Cdc42 itself can transit through the cytoplasm without GDI. For example, Das et al. (2012, Nature Cell Biology 14, 304−312) show using FCS that the rapid−mobility fraction of Cdc42 (presumably freely diffusing protein) drops only about 2−fold in *rdi1* mutants compared to wild−type cells (see their Figure 1). That is, there is still a substantial pool of cytoplasmic Cdc42 in cells lacking the GDI. In biochemical assays involving the exchange of Cdc42 from one liposome to another, Johnson et al. (2009, J Biol Chem 284, 23860−23871) also concluded that Cdc42 comes off membranes with or without the GDI, and that the main effect of the GDI is to stabilize Cdc42 in the aqueous phase. Thus, it seems most likely that Cdc42 exchanges between membrane and cytoplasm readily (if slowly) without the GDI. We speculate that binding interactions of GTP−Cdc42 make it much less prone than GDP−Cdc42 to come off the membrane. We now include some of this discussion in a newly added paragraph (fifth paragraph of the subsection “Testing the competition model: reducing polarity protein mobility”).

*Because of its enhanced membrane binding, the Bem1-CAAX allele certainly also has effects on the levels of available Bem1 (even if global levels are unchanged) and on Bem1 diffusion rates at the membrane. I thus do not think that this approach allows them to specifically slow down membrane-cytosol exchange, nor make any strong conclusion about it.*

It is not clear to us that diffusion of Bem1−CAAX at the membrane would be different from that of Bem1, as the majority of Bem1 at the membrane is expected to be associated (directly or indirectly) with Cdc42, which has the same prenyl tail.

However, we do agree that Bem1−CAAX association with the membrane may translate to less “available” Bem1, as peak Bem1−CAAX levels at the polarity site are lower than peak Bem1 levels (Figure 4). To ask whether reducing available Bem1 might similarly slow competition and cause multi−budding, we have now examined *BEM1/bem1Δ* (and *CDC24*/*cdc24Δ*) hemizygous strains. Except for a lower incidence of multi−cluster intermediates, these cells behave similarly to the wild−type in terms of competition timing and do not produce multibudded cells (Figure 13). We conclude that unlike Bem1−CAAX or Cdc24− CAAX, reducing available Bem1 or Cdc24 does not dramatically slow competition. On the other hand, overexpression of Bem1 does slow competition (Howell et al. Cell 2009), though nowhere near as much as Bem1−CAAX.

Author response image 1.Competition in *BEM1/bem1Δ* and *CDC24*/*cdc24Δ* hemizygotes.(**A**) Coexistence time for multi−cluster intermediates is plotted (each dot is one cell) for strains of the indicated genotypes. (**B−D**) Inverted maximum−projection strips showing examples of competition in wild−type (**B**), *BEM1/bem1Δ* (**C**), and *CDC24*/*cdc24Δ* (**D**) strains. Arrows indicate multiple clusters, and after competition the winning cluster is indicated by a black arrow while the position(s) of the losing cluster(s) are indicated by fainter grey arrow(s). The polarity marker is Bem1−GFP in all cases.**DOI:**
http://dx.doi.org/10.7554/eLife.11611.027

Reviewer #1 (Minor Comments):

*Most or all of the study appears to be conducted in an* rsr1∆ *background, though this is not stated. This should be made explicit.*

Agreed. This is now explicitly stated and explained, first paragraph of Results.

*The first part of the manuscript investigates candidate proteins that may be involved in stabilizing the polarity patch, and concludes that neither actin nor septins appear to have this function. Though I would agree with this conclusion, the description of the septin data seems inaccurate. In Video 4, there is a weak Cdc3 signal on the disassembling left patch at timepoint 8:15, and one before the transient disassembly of the right patch at 9:45 to 11:15, in agreement with recent data that septins contribute in negative feedback for Cdc42 patch disassembly (see for instance Okada et al. Dev Cell 2013). Thus, septins may well contribute to patch destabilization in addition to any competitive mechanism.*

Agreed. We have reworded the text (in the fourth paragraph of the subsection “Testing candidate stabilizers”) to acknowledge that septin signals are sometimes visible at both clusters during competition. As pointed out by the reviewer, that does not affect our conclusion that they do not act as stabilizers. We also discuss and cite Okada et al. 2013 in this context.

*The data on* rdi1∆ *slowing down Cdc42 exchange and producing multi-buds reproduces previously published data (see Slaughter, Dev Cell 2009; Freisinger, NatComm 2013). This data could be provided as supplement.*

We agree that the data on slowing exchange of Cdc42 in *rdi1* mutants is largely confirmatory, but as we understand, the preferred *eLife* policy is to have the main data in the primary figures, so we did not include a supplement.

*I am bothered by the fact that the authors explain at the beginning of their manuscript that GFP-Cdc42 is poorly functional (and thus justify use Bem1-GFP instead), but use this allele to quantify Cdc42 turnover without any disclaimer. In Figure 1, they use a Cdc42-mCherry^SW^ allele modeled on a recently published functional allele in* S. pombe *(Bendezu et al., PLoS Biol 2015; which should be cited), which they show in Figure 9 is not fully functional in* S. cerevisiae*. This information should be better integrated in the text. As it stands, Figure 9 is only referenced in the Materials and methods section (and this also applies to Figure 10). Also, the Cdc42-mCherry^SW^ strain construction is not entirely clear. This construct is said in the Materials and methods to be integrated at the URA3 locus (under which promoter is not stated), but is said in the legend to Figure 9 to replace endogenous cdc42. This should be clarified.*

We apologize for the lapse and have revised the figure legend to make these issues transparent. The construct is expressed from the *CDC42* promoter but integrated at *URA3*. It “replaces” endogenous *CDC42* in the sense that the strain carries a deletion of the endogenous *CDC42*, but the construct is at *URA3*. We have moved this figure (new Figure 1) and now cite it when discussing probe functionality at the beginning of the Results.

*The reasons for the poor viability of the Cdc24-CaaX allele are unclear. Previous work showed that a myristoylatedallele of Cdc24, also targeted to the membrane, grows well even in absence of* rsr1 *(Shimada et al., EMBO 2004).*

Shimada et al. 2004 expressed Myr−Cdc24 on top of the endogenous Cdc24, so they never checked functionality in the same way we do (i.e. replacing the endogenous Cdc24). Myr−Cdc24 localized quite poorly to the plasma membrane, so it does not appear to be as tightly membrane associated as Cdc24− CAAX.

*The Bem1-Caax localization in Figure 4 looks very different from Figure 3. Why?*

Figure 3 (now 4E) shows a single medial−plane slice of the cells, imaged on a scanning confocal. Figure 4 (now 6C) shows maximum−intensity projections of 30−image z−stacks, taken on a spinning−disk confocal.

*I am confused by the limiting or non-limiting nature of polarity factors and the statements made around this by the authors. They first state that "the ability of cells to polarize simultaneously in up to four directions without any overexpression indicates that polarity factors and other components required for budding are present at higher levels than needed to make one bud". I am not sure I understand how competition works if the substrate(s) for competition is not limiting. In the multi-site, multi-budded cells, is the local abundance of the polarity factors reduced compared to the single-site, single-budded situation?*

We apologize for the lack of clarity in our presentation. The polarity protein amounts that need to be clustered to specify a successful polarity site may be quite small, so that a cell has enough polarity factors to form several buds, and could do so if each site were content to amass only as much as they needed. But instead polarity sites are insatiable due to positive feedback, and the “winner” of the competition continues to grow until the cytoplasmic pool of polarity factors is sufficiently depleted to stop further growth. At this point, there are insufficient cytoplasmic levels to support another cluster.

We have added a section, “Substrate depletion and negative feedback”, that deals with some of these questions, including the reviewer’s query regarding the levels of polarity factors in each site in multi−site cells. This issue is complicated by the presence of negative feedback: as published elsewhere (Howell et al. Cell 2012, Kuo et al. Current Biology 2014), the winning peak proceeds to disperse polarity proteins and its content can oscillate after it has won the competition. The negative feedback involves Cdc24 phosphorylation. To address the reviewer’s question, we crippled negative feedback by replacing Cdc24 with the mostly nonphosphorylatable Cdc24_38A_, and we deleted *RDI1* so that cells would sometimes make two buds. Then we quantified the amount of Bem1−GFP in the polarity patch immediately before bud emergence, for cells that made either just one bud or two buds (new Figure 10). Despite considerable cell−to−cell variability, the results show that two−site cells indeed have less Bem1 in each site than one−site cells.

*I am further confused by how the statement "because of positive feedback, stochastic activation of a small amount of Cdc42 somewhere on the membrane leads to further accumulation of active Cdc42 until depletion of the cytoplasmic pools of polarity proteins stops the process" can be reconciled with the statement in the next paragraph that "the uniqueness of the Cdc42 peak is not due to insufficient Cdc42 or Bem1-Cdc24". Does the drop in cytosolic Bem1 after competition resolution predicted in Figure 7 really happen? This should correlate with a local increase of Bem1 at the membrane. But Figure 3 showed an overshoot of Bem1 during competition and lower level after.*

Please see the answer to the previous question. The more complex situation in cells is due to negative feedback.

*One conclusion from the data is that the reduced competition observed in Bem1-CaaX and* rdi1∆ *conditions does not qualitatively alter the model, but quantitatively changes parameters and thus the timescale of competition. This would suggest that the multi-budding phenotype could be rescued by delaying G1-S transition until resolution of the polarity patches to a single one. Is that the case?*

Good question. Unfortunately we do not understand what determines the approximately 10 minute interval between initial polarization and bud emergence. Blocking G1ƒS (e.g. with HU) does not affect the timing of polarity or bud emergence, and blocking Start (ClnƒCDK activation) would affect both polarization and budding. We do not know how to delay budding without affecting polarity establishment. We are working on this question, but cannot yet do the requested experiment.

*In the final part of the manuscript, the authors aim to resolve a difference between the output of their model, which yields multiple initial clusters resolving into one, and the model from Klunder et al., 2013, which showed a single initial broad peak sharpening over time, and conclude that the differences are due to parameter value choices. This is an interesting reconciling finding, but their choice of parameters for Cdc42 lateral diffusion is low. The value chosen, 0.0025µm2/s was measured for transmembrane proteins. More recent values for Cdc42 were obtained, ranging from 0.013 µm2/s to 0.0061 µm2/s in* S. cerevisiae *(Marco et al., Cell 2007 and Freisinger et al., NatComm 2013) and up to 10-fold higher using functionally tagged Cdc42 in* S. pombe *(Bendezu et al., PloS Biol 2015).*

The question of the membrane diffusion constant for Cdc42 is a difficult one. The estimate from Marco et al. involved fitting FRAP data on the assumption that CAAX−containing proteins cannot exchange between membrane and cytoplasm. But as discussed above, we believe that assumption to be wrong. Thus, the fit would overestimate the diffusion constant because it doesn’t take “hopping” on and off into account. The Bendezu work is much more convincing, because the shape of the cell permits bleaching of a larger area and GDP−Cdc42 diffuses much more rapidly in the membrane so the measurement is not confounded by the relatively slower exchange (which unlike Marco they do incorporate in generating their fits). But, as they point out, their measurement refers only to GDP−Cdc42:GTP−Cdc42 diffusion was too slow to measure. Of course, there is also no guarantee that the two yeasts would necessarily have a similar diffusion environment as their cell walls are quite different. The relevant (slow) diffusion constant in our model applies to GTP−Cdc42, since GDP−Cdc42 already has high mobility due to its ability to exchange into the cytoplasm. We admit that we do not have a valid measure for this, but we note that no model with rapid diffusion of GTP−Cdc42 would be able to explain how polarity sites in such close proximity (e.g. Figure 6, Figure 7 cell 2, or especially Figure 7 cell 2) could avoid merging together. With specific reference to our re−analysis of Klunder et al. 2013, we note that our main conclusion holds even if we keep their higher diffusion constant, once the volume correction is applied.

*Finally, given most the experiments are done modifying Bem1 properties, it would be necessary to test the effect of limiting Bem1 levels and thus constructing Bem1 hemizygote diploids.*

We have now constructed the *BEM1* hemizygote diploids: the frequency with which we observe multi−cluster intermediates was reduced (by approximately 33%, based on Bem1−GFP movies), while resolution of the intermediates occurred normally (Figure 13). This is in contrast to Bem1−CAAX, where multi−cluster intermediates were morefrequent and took much longer to resolve. Thus, we would argue that the slowed competition is due to slower trafficking of Bem1−CAAX rather than reduced availability.

Reviewer #2:

*Wu et al. test two alternative hypotheses on the regulation of the establishment of a unique polarity site in budding yeast cells. They conclude that a competition between different sites rather than a stabilizing factor is responsible for the collapse of all but one polarity zone. They test the corresponding mathematical models of the two hypotheses and find that their earlier published model explains the underlying mechanism. Results of Figure 5 are convincing that a growing bud can be outcompeted and turned off, disproving the stabilizer hypothesis. The comparison of the computational models supports this conclusion, but does not go beyond earlier explanations of the system*.

We would disagree with that characterization. In a sense, the Segel and Jackson paper cited below did not “go beyond” Turing (1952), but the paper was hugely valuable in that it explained *why* such systems behaved as they did. Although at a much more modest level, our analysis of the Goryachev model goes beyond the simple statement that it can display competition, by dissecting the contributions of different processes and explaining why the peaks compete.

*The conclusion that the competition mechanism works through positive feedback loops without an inhibitor tells me that it works with substrate-depletion. The sentence in the second paragraph of the subsection “Mechanism of competition in a computational model” does not support this idea, but as Bem1 overexpression can also lead to multi-budded cells and its level goes down on Figure 7 assume Bem1 is (at least part of) the limiting substrate in this Turing-like system. If we follow the theory on such systems (Segel, L. A., and Jackson, J. L. (1972) Journal of Theoretical Biology, 37(3), 545-559), then not only changes in diffusion rate, but also changes in the size of cells could have similar effects as the critical wavelength of the underlying Turing mechanism depends on both. Thus it would support the conclusions if the authors could show that cells with larger size (e.g. HU block and release) can form multiple peaks, especially if combined with the slow diffusing membrane bound proteins. From their current data they might be already able to analyze if there is a correlation between cell size and number of appearing buds in the strains investigated in Figure 6.*

The reviewer is correct with regard to substrate depletion, and we have added a new section to emphasize that point. The sentence was intended to convey the idea that the total amount of polarity factors suffices for several buds, which does not contradict the substrate depletion idea, but we concede that the text was unclear and have deleted that sentence and re−written to try to make the issue clearer, as discussed above.

Stability analysis as in Segel and Jackson only refers to the system very close to the homogeneous steady state: once we leave that regime (and competition occurs well into the nonlinear regime), stability analysis cannot predict how many peaks will survive to the polarized steady state. It is true that a precondition for >1 peak is instability to higher wave numbers (which is why Klunder et al. appeared to provide a challenge to the competition hypothesis), but beyond that, stability analysis does not predict anything about how many peaks there will be.

With regard to the expectation that larger cells would develop more (stable) clusters, Otsuji et al. (2007, PLoS Comp Biol 3, 1040−1054) showed that a mass−conserved Turing system (of which ours is one example) could in principle yield a single−peak final steady state regardless of size. They did not determine what it is that causes some models to behave this way while others don’t, and it is one of our goals to elucidate that, but we aren’t there yet.

Experimentally, we have not noticed a consistent correlation between cell size and the number of buds, but it may be we simply have not reached a large enough size.

*If it is a competition mechanism with positive feedbacks in a Turing-like system then a gradient of the limiting compound could be showed. Probably by expressing in low level, an extra copy of Bem1-GFP could show an actual gradient around polarity zones and the change in size of these zones during competition.*

The gradient in the cytoplasm predicted by modeling is minuscule (<0.1%), because of rapid cytoplasmic diffusion. There is also a technical issue that would preclude detection of dips in cytoplasmic concentrations under each membrane peak: diffraction of light from the probes concentrated in the peak at the membrane would mask the dip in the cytoplasm. We apologize if we misinterpreted this question.

*The authors chose to drop the negative feedback loop from their last model (Kuo et al. 2014) and replace with a constant high GAP activity. This indeed allowed them to explain the issue with the uniqueness of the final peak, but does not explain how the single peak disappears sometimes. Furthermore, the choice of dropping a potential inhibitor restricted the Turing-type of explanation to the substrate-depletion model. Probably both work better than the stabilizer model, but I feel this as a step back from their earlier results. Indeed it is not clear what new we learned from the model compared to earlier analysis of various versions of it.*

We believe that we have learned two things.

1) We provide the first explanation as to why the model shows competition (the fact that it does so was known, but we dissect the basis by breaking the process down to its parts). This was not obvious (to us, at least) from just looking at the rather complex model.

2) We explain why the Klunder et al. model did not allow for competition, and account for the discrepancies between the models.

We completely agree that negative feedback is necessary to explain features like single−peak disappearance, but that is not the focus of this study. Mutants that reduce negative feedback considerably (*CDC24*^38A^) still show competition, with dynamics similar to wild−type (Kuo et al. 2014). In addition, we now show that mutants with reduced negative feedback still generate two−budded cells when exchange of Cdc42 is slowed (new Figure 10). Thus, we believe that negative feedback is unlikely to be central to the process of competition. We also note that a similarly−parameterized model that includes negative feedback displays competition behavior that looks quite similar to that in the model lacking negative feedback (Figure 14).

Author response image 2.Competition in a model with negative feedback.(**A**) Snapshots of competition from two−peak unstable steady state. (**B**) Cytoplasmic protein levels during competition. (**C**) Flux plot as in Figure 9. Parameters were as in Kuo et al. 2014. Overall behavior is very similar to that shown in Figure 9 for model with no negative feedback.**DOI:**
http://dx.doi.org/10.7554/eLife.11611.028

However, as reported in Howell et al. 2012, models with negative feedback can, with different parameter choices, allow coexistence of polarity clusters instead of competition. This makes the full model very complex to analyze, and we prefer to defer that analysis until we get a better understanding of the factors that can eliminate competition to allow coexistence in models with negative feedback.

[Editors' note: the author responses to the re-review follow.]

Reviewer #1 (Minor Comments):

*The authors cite the paper by Otsuji et al. claiming that any initial pattern could end up with a single peak in a substrate-depletion reaction system, but as they correctly also add, this was shown only for mass-conserved Turing systems. They claim that their system is mass-conserved, which might be true for the time window they investigate the cells (< 15 mins). The simulations were initiated with identical perturbations on two sites and followed what steady state is reached. To further validate the limiting factor idea it would be interesting to test what happens as the cells grow. There are several papers (by Crampin and Maini) on the behaviour of the Turing pattern in growing domains. I was wondering how the proposed model would behave if growth is considered.*

This is certainly an interesting question, which we are beginning to investigate. However, we do not yet know what general rules apply (if any) in growing domains, and we believe that this is an issue that will require considerable work to resolve, and lies outside the scope of the present manuscript.

*A related question could be if and how the model could explain the periodic budding of clb1-6del cells. Could the increase in the concentration of the limiting factor induce a second tip to assemble?*

*In general the point I would like to see (at least by the model) is that the second tip can be formed not only for slowed down competition, but also if the second zone is above a given distance from the first or the limiting factor is not limiting anymore. These could further prove the limiting factor hypothesis.*

The clb1-6del cells are really interesting, but not directly related to the issue of competition. Basically they always have one polarity/growth site like wild-type cells (at least for the first few cycles): it is just that after growing in one direction for a while the site shifts back to the mother and starts a new bud, repeatedly. Why the site switches is unclear, though Haase and colleagues argue that it is driven by an internal transcriptional oscillator.

Reviewer #2:

*My main concern, which was that addition of a prenyl group would lock the proteins on the membrane, has been largely addressed, but I would encourage the authors to slightly extend their work on Cdc24, as suggested below. For Bem1-CAAX, the anchor-away experiment is convincing and indeed shows that Bem1-CAAX is mobile on and off the membrane. I would suggest that, in the flow of the paper, it would be better to describe this at the top of the Bem1-CAAX allele description as a way of introducing it as a tool to reduce exchange dynamics, rather than at the end. The conclusions made in the fourth paragraph of the subsection “Testing the competition model: reducing polarity protein” would then be more straightforward. In these lines, I would also suggest rephrasing the meaning of FLIP and FRAP quantification, as these do not directly show membrane-cytosol exchange dynamics, because lateral diffusion will also be a contributing factor. The proposed re-organization would also help in understanding the next paragraph, in which the logical link between the slowed membrane-cytosol exchange of prenylated protein and the previous analysis of Cdc42 dynamics is not entirely clear.*

We have reorganized the text and reordered Figure 4 and Figure 5 as suggested. In addition we have rephrased the interpretation of FLIP and FRAP as suggested. We agree that this flow is more intuitive and thank the reviewer for the suggestion.

*By contrast, the localization of Cdc24-CAAX to the nuclear membrane in haploid cells does not imply a detachment from the membrane. The nuclear membrane is connected to the ER, and it is possible that the relocation occurs as Cdc24-CAAX remains bound to the membrane rather than transit through the cytosol. Thus, the manuscript has been significantly improved and now convincingly shows that Bem1-CAAX shuttles between membrane and cytosol, but the same level of confidence does not apply to Cdc24-CAAX, as it would be predicted to be nuclear, rather than at the nuclear periphery in haploids if that were the case. I would feel much more comfortable if the data was confirmed using an alternative mode of binding these proteins (or at least Cdc24) peripherally to the plasma membrane, for instance through an amphipathic helix, a C2 or a PH domain.*

We agree that localization of Cdc24-CAAX to the nuclear membrane could in principle occur without detachment from the membrane (though the fact that the integral membrane Cdc24-Snc2 does not go to the nuclear membrane suggests it actually does involve detachment). We have removed that data and argument from the paper. However, it seems to us logical that if the polybasic-prenyl motif doesn’t anchor Bem1-GFP to the membrane, it wouldn’t anchor any other cytoplasmic protein (like Cdc24) either.

With regard to amphipathic helix, C2, or PH domains: these are already present in the endogenous polarity proteins – Cdc24 has a PH domain, Bem1 has a PX domain, and Cdc42 effectors (Ste20, Cla4, Gic1, Gic2, Boi1, Boi2) that bind Bem1 all have amphipathic helices or PH domains. These are thought to provide very weak but essential membrane binding domains (Takahashi and Pryciak, 2008., Mol Biol Cell 18: 4945-4956). Because they are so weak, we believe that adding them to Cdc24 would be unlikely to slow dynamics significantly enough to have a big effect on competition.